# Evidence of Learned Look-Ahead in a Chess-Playing Neural Network

**Erik Jenner**[1]
UC Berkeley

**Shreyas Kapur**
UC Berkeley

**Vasil Georgiev**
Independent

**Cameron Allen**
UC Berkeley

**Scott Emmons**
UC Berkeley

**Stuart Russell**
UC Berkeley

## Abstract

Do neural networks learn to implement algorithms such as look-ahead or search "in the wild"? Or do they rely purely on collections of simple heuristics? We present evidence of *learned look-ahead* in the policy and value network of Leela Chess Zero, the currently strongest deep neural chess engine. We find that Leela internally represents future optimal moves and that these representations are crucial for its final output in certain board states. Concretely, we exploit the fact that Leela is a transformer that treats every chessboard square like a token in language models, and give three lines of evidence: (1) activations on certain squares of future moves are unusually important causally; (2) we find attention heads that move important information "forward and backward in time," e.g., from squares of future moves to squares of earlier ones; and (3) we train a simple probe that can predict the optimal move 2 turns ahead with 92% accuracy (in board states where Leela finds a single best line). These findings are clear evidence of learned look-ahead in neural networks and might be a step towards a better understanding of their capabilities.

## 1 Introduction

Can neural networks learn to use algorithms such as look-ahead or search internally? Or are they better thought of as vast collections of simple heuristics or memorized data? Answering this question might help us anticipate neural networks' future capabilities and give us a better understanding of how they work internally. Recent work has found interesting cases of learned optimization or reasoning in neural networks (von Oswald et al., 2023a,b; Akyürek et al., 2023; Brinkmann et al., 2024). However, these works focus on simple algorithmic domains, with models that are trained specifically for those research purposes. Instead, we ask: what computations do networks learn to perform "in the wild" in more complex domains?

We study this question in the microcosm of chess. Neural networks are surprisingly strong at chess, arguably approaching grandmaster level (Ruoss et al., 2024)—how do they achieve that performance? Both reasoning and heuristics are plausible mechanisms. For example, human players and manually designed chess engines perform *look-ahead*—they reason about which moves they will make *in the future*. On the other hand, a network might simply learn heuristics based on the *current* state, such as playing knight "fork" attacks—which are often advantageous—purely based on what they look like geometrically. There is evidence that residual networks (such as transformers) tend to additively aggregate results from many shallow circuits (Veit et al., 2016) and incrementally improve their predictions (nostalgebraist, 2020; Belrose et al., 2023; Din et al., 2023), lending credence to this idea that networks might implement a large collection of heuristics. Since we know how to

---

[1]Corresponding email: `jenner@berkeley.edu`

38th Conference on Neural Information Processing Systems (NeurIPS 2024).

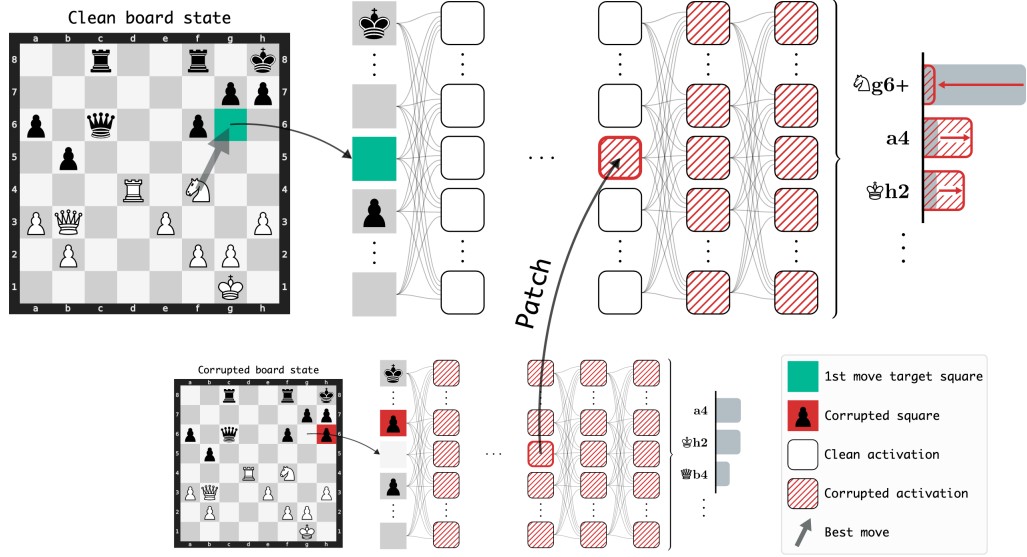

Figure 1: *Activation patching* lets us study where important information is stored in Leela. Here, we patch an activation in one particular square and layer from the forward pass on a "corrupted" board state (bottom) into the forward pass on a "clean" state (top). Each row in the network corresponds to one chessboard square, which Leela treats like a token in a language model. The intervention drastically affects Leela's output (right), telling us that the activation on the patched square stores information necessary for Leela's performance in this state. Only patching on specific squares has significant effects. See https://leela-interp.github.io/ for more (animated) examples.

hand-design chess engines, we know what reasoning to look for in chess-playing networks. Compared to frontier language models, this makes chess a good compromise between realism and practicality for investigating whether networks learn reasoning algorithms or rely purely on heuristics.

In this work, we look for evidence of look-ahead in the policy and value network of Leela Chess Zero (Leela Chess Zero team), or Leela for short. Leela is an MCTS-based system (like AlphaZero (Silver et al., 2018)) and the strongest deep neural chess engine (Haworth and Hernandez, 2021). We use only its policy/value network, without external search, since we want to study algorithms that emerge *within* the network. Even the policy network, with only one forward pass per state, reaches a rating over 2600 on Lichess (Appendix B) and is at least as strong (lepned, 2024) as a recent model from Ruoss et al. (2024).

Our inroad for interpreting Leela is that it is a transformer that treats each chessboard square like a token in a language model. Thus, we can consider activations on specific squares or attention weights between squares. We consistently find that these activations correspond to their squares in meaningful ways; for example, information about a move typically seems to be stored in activations on the squares involved in that move. This lets us apply common interpretability techniques to Leela.

We show that Leela has learned to use look-ahead in certain states. Leela internally represents future moves of the optimal line of play, and these representations are causally important for Leela's output. We present three lines of evidence. (1) Activations on the target square (where the piece lands) of certain *future* moves have a substantially outsized impact on the network's output, as determined by activation patching (Fig. 1 and Section 2.3). (2) We identify attention heads that move information "forward and backward in time." For example, one attention head often moves crucial information from the target square of a future move to the target square of an earlier move. (3) A simple, bilinear probe (Hewitt and Liang, 2019) on a subset of Leela's activations can read off the best move two turns into the future with 92% accuracy.

Our contributions are: (1) We give evidence that neural networks can learn algorithms involving look-ahead "in the wild." (2) We take first steps toward a mechanistic understanding of how look-ahead might be implemented in Leela to help it play chess. (3) We introduce techniques that might be useful

for interpretability more generally (e.g., using a weaker model to automatically generate corruptions for activation patching, as we will explain in Section 2.3). Our code is available on Github.

## 2 Experimental Setup

This section will describe the model, dataset, and techniques we use. The following section will describe the specific experiments and their results. We ran all experiments on an internal cluster. Each experiment takes at most a few hours on a fast GPU (e.g., an A100) and about a day for all experiments combined.

### 2.1 Leela Chess Zero

Leela is a chess engine based on Monte Carlo Tree Search (MCTS), similar to AlphaZero (Silver et al., 2018). We focus solely on its policy network, which takes a single board state as input and outputs a probability distribution over all legal moves. For ease of exposition, we ignore the value network in the main paper, but we show in Appendix C that our results also apply there (likely because the two networks share a common body). When we say "Leela" elsewhere in this paper, we mean the network rather than the full MCTS system.

The key to understanding our analysis is that Leela is a transformer that treats each of the 64 chessboard squares as one sequence position, analogous to a token in a language model. This means each square has its own representation in the embedding space, allowing us to analyze activations and attention patterns on specific squares. Unlike a causal language model, attention is bidirectional between squares; there is no autoregressive prediction. Leela has 15 layers and 109M parameters, about the size of GPT2-small (Radford et al., 2019).

The network computes a logit for every possible move, corresponding to moving a piece from one square (the source) to another (the target). Each logit is computed using only the final embeddings at the source and target square. There are several peculiarities of Leela's architecture that aren't crucial for understanding our results, so we discuss them in Appendix A.

### 2.2 Puzzle dataset

To study look-ahead, we need a dataset of board states where Leela is especially likely to use look-ahead. We are *not* claiming that Leela uses look-ahead in *every* state. Even human players can analyze many states heuristically if there are no complex tactical considerations that require explicit look-ahead. Thus, we focus on complex states that are likely difficult to evaluate heuristically.

As a starting point, we use 900k puzzles from the Lichess puzzles dataset (Lichess team). Each puzzle has a starting state with a single winning move for the player whose turn it is. It is also annotated with the principal variation, the optimal sequence of moves for both players from the starting state. All puzzles we consider have at least three moves[2] in their principal variation, see Fig. 2 for an example.

We discard puzzles that a smaller and weaker version of Leela can solve. This ensures the states in our dataset are challenging and more likely to require look-ahead. We further filter for puzzles that Leela solves correctly, simply so that we can apply our interpretability methods: all our methods check whether Leela internally represents a specific future line. In correctly solved puzzles, we can look for representations of the correct continuation, but for puzzles that Leela fails to solve, it's unclear which line we should look for. Leela may be representing an incorrect continuation (and hence fail the puzzle), but there are many incorrect lines (vs only one correct one). After this filtering process, 22.5k puzzles remain (mainly because many of the original puzzles are easy enough to be solved by the smaller model, which means we discard them). See Appendix D for details.

The filtering process leads to an overrepresentation of states where the 1st and 2nd move (i.e., the opponent's response) have the same target square. The 1st and 2nd target square coincide in 83% of the filtered puzzles, compared to 47% on the original dataset. This is because the starting player often sacrifices a piece that the opponent captures. These puzzles involving sacrifices may be more difficult for the weak model and thus overrepresented in our dataset. Interestingly, the results we present in

---

[2]Note for chess players: for simplicity, we use the term "move" for what's often called a "ply" or "half-move". We never mean a "full move", i.e. a move by both white and black.

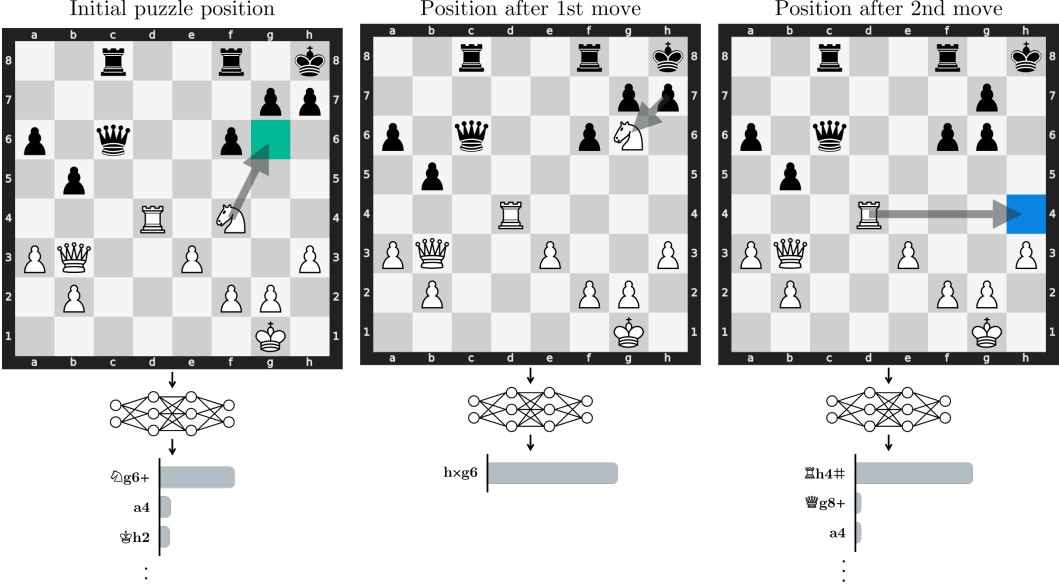

Figure 2: *Top row:* An example of the puzzles we use. It is white's turn in the starting state, and the only winning action is to move the knight to **g6**. Black's only response is taking the knight with the pawn; then white checkmates by moving the rook to **h4**. We will see the colored squares again: the target square of the 1st move in this *principal variation* (**green**) and the target square of the 3rd move (**blue**). *Below:* Leela receives each state as a separate input and computes a policy in that state.

Section 3 are weaker on puzzles where the 1st and 2nd move target square differ and are stronger in cases where they coincide. Perhaps sacrifices are inherently "more interesting" for Leela, or some model components we identify are specialized for dealing with sacrifices. See Appendix H for results on both subsplits of the data; in the main text, we always present results on all puzzles.

## 2.3 Activation patching

Activation patching[3] is a technique for measuring the causal importance of specific model components. For any given board state and model component (such as a particular square in a particular layer), it can tell us how important that component is for producing Leela's output in that state.

To do so, activation patching replaces the activations of the component in question with those from a different forward pass (Fig. 1). For a given "clean" board state from our dataset, we use a "corrupted" version of that state with a small modification, such as a piece being added or removed. In the version of activation patching we use, we run a forward pass on the clean state, but "patch in" activations from a forward pass on the corrupted state at the component we're analyzing. We then continue the forward pass as normal. If this intervention changes Leela's output significantly (compared to a clean forward pass without intervention), the patched component must have contained *necessary* information about the clean state that differed in the corrupted state.

The choice of corruption determines which model mechanisms we can study with activation patching. If the corrupted state is very different from the clean state, many model components will have different activations, and activation patching won't tell us anything specific about look-ahead. Instead, we want a corrupted state that is similar to the clean one but differs in some key detail that has an outsized effect on what move is best. Look-ahead or other sophisticated algorithms should pick up on the importance of this difference, but shallow heuristics should mostly ignore it.

To automatically find such "interesting" corruptions, we again use a smaller and weaker version of Leela. We generate many small random corruptions, each modifying only a single square or piece position. Then, we select corruptions that have a large effect on Leela's preferred move but

---

[3]Similar (Heimersheim and Nanda, 2024; Zhang and Nanda, 2024) to causal mediation analysis (Vig et al., 2020), causal tracing (Meng et al., 2022), or interchange interventions (Geiger et al., 2021)

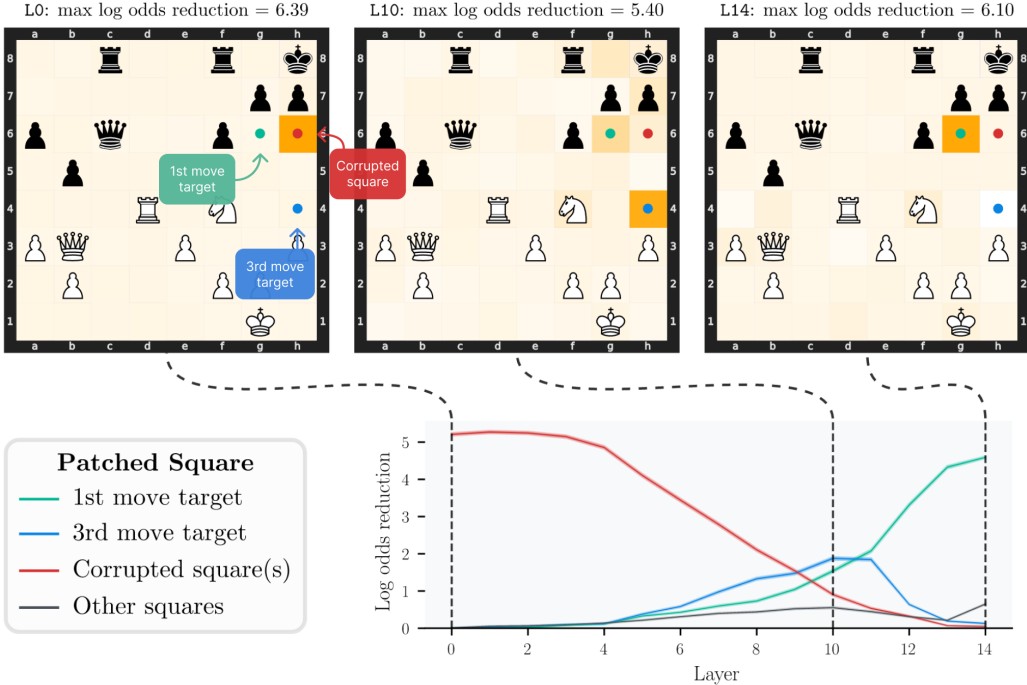

Figure 3: Results from activation patching in the residual stream. The top row shows results in a single example state at three select layers. Darker squares correspond to larger effects from intervening on that square. In the early layer, the effect is strongest when patching on the **corrupted square h6**, then in middle layers, the **3rd move target square h4** becomes important, and finally the **1st move target square g6** dominates in late layers. The line plot below shows mean effects over the entire dataset, demonstrating that this pattern holds beyond just this example. The "**other squares**" line is the *maximum* effect over all 61 other squares (where the maximum is taken per board state and then averaged). Error bars are two times the standard error of the mean.

a small effect on the weaker model's output. This targets mechanisms that explain why Leela gets these puzzles correct while the weaker model does not, making them more likely to be related to look-ahead. The exact algorithm for generating and filtering corruptions is described in Appendix E.

# 3 Results

If Leela uses look-ahead, it needs internal representations of future moves. These could be located anywhere in the model, but we will argue for a specific hypothesis: that Leela represents future moves on their source or target squares. This hypothesis is motivated by the fact that at the end of the network, the logit of a move depends only on its source and target square. We are guessing that something similar holds within the network for *future* moves, and our results bear out this hypothesis.

We present three lines of evidence for this specific look-ahead hypothesis. First, we show that activations on the target square of the move two turns into the future are unusually important for Leela's output. Second, we find attention heads that appear to help Leela consider the consequences of future moves, as well as a head that moves information "backward in time." Third, we demonstrate that a simple probe can predict the optimal move two turns into the future with 92% accuracy. The convergence of these three lines of evidence strongly suggests that learned look-ahead is an important mechanism behind Leela's impressive performance on our dataset of puzzles.

### 3.1 Activations on future move squares are unusually important

If Leela uses look-ahead and represents future moves on their source or target squares, then we expect activations on these future squares to be especially important for its output. We test this using activation patching: we corrupt activations one square and layer at a time by patching in activations from the corrupted puzzle into the clean forward pass. We measure the causal effect of an intervention by the change in log odds assigned to the ground-truth best move. If Leela is using look-ahead, we expect especially big reductions in log odds when we patch on the squares of optimal future moves.

Indeed, we find that patching on the **target square of the 3rd move** reduces model performance by an unusual amount (Fig. 3). In layer 10, this intervention reduces the log odds of the correct move by an average of $1.88 \pm 0.04$ ($2\sigma$ standard error of the mean), which corresponds to a reduction in probability from e.g. 50% to 13%.

Patching on the **corrupted square** or the **1st move target square** also has large effects. This is unsurprising: the corrupted square is the only difference in the input encodings, so in early layers, this square is responsible for any subsequent differences in the forward passes and output. The 1st move target square directly affects the logits of the correct move, so has a big influence in late layers.

Patching on any **other square** has much smaller effects. For each puzzle, we consider the *biggest* effect from patching on any square other than the 1st move target, 3rd move target, or corrupted square, and average those maxima over puzzles (this is the "**other squares**" line). Even though this is a maximum over 61 squares, the effects are much smaller than those for the 3rd move target. For example, in layer 10, the mean of these maximum log odds reductions is only $0.55 \pm 0.01$, compared to the $1.88$ for the 3rd move target. This shows that information stored on the 3rd move target square is unusually important for Leela's output, more so than information on most other squares.

We are unsure why the squares of the 2nd move aren't similarly important. This may simply be because the opponent's move is typically "obvious" in our dataset or because suppressing the opponent's best response doesn't reduce the quality of the 1st move. We confirm in Appendix H that this is not just an artifact of the overlap between 1st and 2nd move targets. Similarly, we don't know for certain why Leela seems to mainly store information on target squares rather than source squares (both for the 3rd move and for the immediate 1st move).

### 3.2 Attention heads move information forward and backward in time

We have seen that the target square of the 3rd move in the principal variation contains unusually important information. If Leela uses look-ahead, this information must somehow inform its decision for the 1st move. An algorithm involving look-ahead might consider the consequences of making the 3rd move and then propagate that information back to earlier timesteps. In this section, we will find evidence of these processes. We identify an attention head that moves information from the 3rd move target square "backward in time" to the 1st move target square, as well as heads that seem to move information "forward in time" to consider the consequences of the 3rd move.

**L12H12 moves information "backward in time"** In the previous section, we found *squares* potentially involved in look-ahead just by measuring their importance using activation patching, so we will try the same simple approach for *attention heads*. When we patch the output of one head at a time from the corrupted to the clean forward pass, one head stands out: L12H12 (the 12th head in the 12th layer) has a much larger average effect than any other attention head (Fig. 4).

So what does this head do? Anecdotally, we noticed that the entry of the attention pattern $Q^T K$ with the 1st move target as the query and the 3rd move target as the key is often large: in 29.8% of puzzles, it is higher than all 4095 other attention entries in L12H12. In other words, it seems that L12H12 often moves information from the 3rd move target "backward in time" to the 1st move target, where it is needed for the immediate decision.

We test this hypothesis more directly by ablating this specific attention entry. Remarkably, zeroing out this single entry reduces the log odds of the correct move by more than 1.5 in more than 10% of puzzles (Fig. 5), corresponding to a reduction in probability from, e.g., 50% to 18%. This effect is much larger than simultaneously ablating all 4095 other L12H12 attention weights.

These results are particularly striking given the scale of the intervention: we are zeroing just one floating-point number out of about 1.5 million attention entries and many other types of activations.

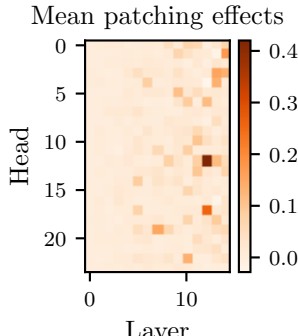

Mean patching effects

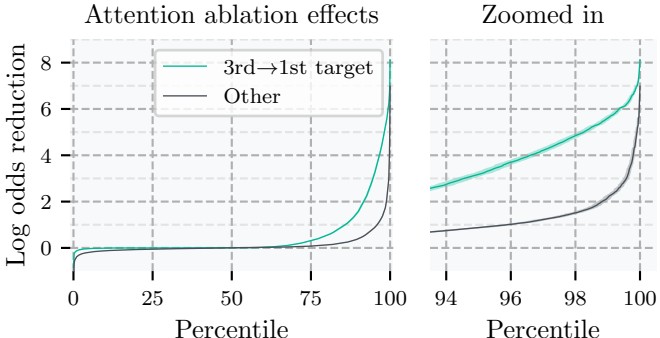

Attention ablation effects    Zoomed in

Figure 4: Mean log odds reduction from activation patching attention head outputs one head a a time. The head that stands out the most is L12H12.

Figure 5: Zero-ablations in the attention pattern in L12H12. *Green line:* ablation of the attention entry with key on the 3rd and query on the 1st target square. *Gray line:* ablation of all 4095 other entries at once. The lines show the effect at a given percentile of puzzles sorted by effect size. Error bars are 95% CIs; see Appendix G.

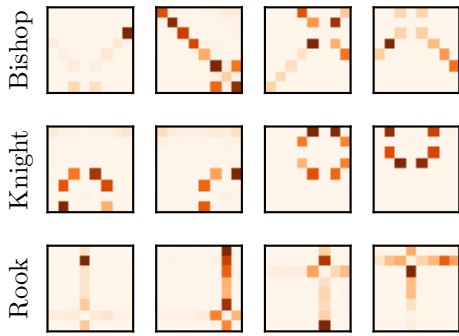

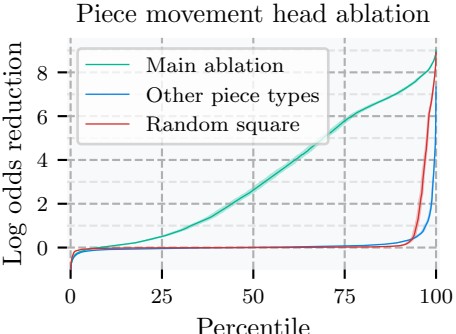

Piece movement head ablation

Figure 6: Attention patterns of random *piece movement heads*. Each row is one head, each column a random puzzle. For each puzzle, we plot the attention pattern for a fixed random query square and varying key squares. A fixed key and varying query give similar results.

Figure 7: Effect of zero-ablating attention entries moving information out of the 3rd move target square, in piece movement heads corresponding to the 3rd move piece type. Error bars are 95% CIs, see Appendix G for details.

That this has such an outsized effect strongly suggests that this specific information pathway is crucial for Leela's decision-making process in a substantial fraction of states. This is consistent with our residual stream patching results, where the 3rd move target was important in earlier layers (around L10 and L11), but after L12, the 1st move target dominates. L12H12 seems to be one mechanism enabling this shift.

**"Piece movement heads" help analyze consequences of future moves**    In addition to L12H12, which moves information "backward in time," we find evidence of attention heads moving information "forward" to analyze the consequences of future moves.

Throughout the network, Leela has several attention heads whose attention patterns closely resemble legal moves for specific piece types (Fig. 6). We call these *piece movement heads*. For example, there are "knight heads" whose attention is focused on squares reachable by knight moves, and similar heads exist for bishops and rooks. By manually inspecting attention patterns in random example states, we identify 22 "knight heads," 27 "bishop heads," and 29 "rook heads" spread relatively evenly throughout the 15 layers (out of 360 heads total in the network). We also found weaker signs of heads for other piece types but exclude those here. King heads are difficult to study because our dataset contains only few examples where the first move is a king move. Pawns move differently for normal

moves vs captures, and it seems those functions are captured by different heads, so we ignore them for simplicity. Finally, there seem to be at most a handful of queen heads, and we suspect that queens are handled mainly by combining rook and bishop heads.

We hypothesize that these piece movement heads are used, at least in part, to analyze the consequences of future moves. If a knight will be moved on the 3rd move (and thus end up on the 3rd move target square), the network might use knight heads to determine what effects a knight on the 3rd move target square would have.

To test this hypothesis, we again ablate information flow out of the 3rd move target square, but this time in piece movement heads instead of L12H12. For each puzzle, we zero out all attention entries with their key on the 3rd move target square. We do this only in piece movement heads corresponding to the piece type of the 3rd move because we want to see whether the network is attending to the consequences of this move specifically. We also do *not* ablate the attention entry in piece movement heads between the source and target square of the 3rd move—we still want to let the network consider the 3rd move itself, just not its consequences. In the running example puzzle from Fig. 2, the 3rd move is a rook to h4, so we would ablate information flow out of h4 in all "rook heads," except to the source square d4. This should prevent Leela from "noticing" the effects that the rook would have on h4 (namely delivering checkmate).

For this experiment, we focus on the subset of our puzzles where the principal variation (as given by the Lichess dataset) is longer than 3 moves. This ensures that there are potential consequences to analyze after the 3rd move rather than having reached an easy-to-evaluate state by then.

The ablation typically has a large effect on the network's output (Fig. 7). In 60% of puzzles, the log odds of the top move are reduced by at least 1.5. In contrast, ablating both other piece movement heads (for different piece types) or ablating on a random square has little impact on performance. This suggests our intervention blocks a very important network mechanism rather than simply reducing performance for generic reasons.

While the heads we've identified seem involved in analyzing future moves, they are certainly not a full explanation of how Leela implements look-ahead. Piece movement heads likely also serve many simpler functions unrelated to look-ahead. Conversely, there might be heads other than L12H12 involved in moving information backward in time, and there are likely many heads involved in look-ahead that don't specialize in one piece type.

### 3.3 Simple probes can predict future moves

We have seen evidence that the 3rd move target square contains information that is unusually important for Leela's output and is moved by attention heads in ways consistent with look-ahead. But can we go a step further and show that this square explicitly encodes information about what the 3rd move *is*?

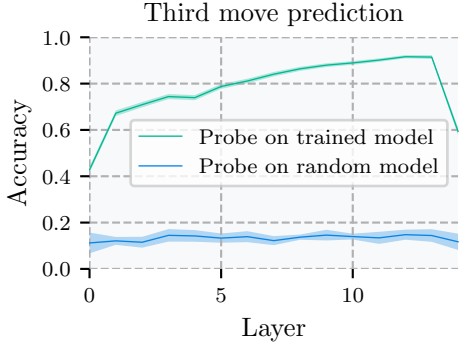

We find that this is indeed possible: probes inspired by our analysis of L12H12 can predict the 3rd move with 92% accuracy. Recall that our dataset only includes puzzles that Leela solves correctly and with a unique principal variation. This makes it possible *in principle* to achieve such high accuracy, but it is nonetheless remarkable that a simple probe can predict a move two steps into the future.

Figure 8: Results of a bilinear probe for predicting the 3rd move target square. Errors combine standard errors of the mean for five probe training runs with standard errors for accuracy estimates; see Appendix G.

The architecture of our probe is directly motivated by our observations on L12H12. Recall that the attention pattern of L12H12 often has a large entry where the 1st move target attends to the 3rd move target. This entry in the attention pattern $Q^T K$ can be written as $h_{t_3}^{11}{}^T W_Q^T W_K h_{t_1}^{11}$, where $h_i^L$ denotes the residual stream activations after layer $L$ on square $i$; $t_j$ is the target square of the $j$-th move; and $W_Q$, $W_K$ are the query and key weight matrices of L12H12. We could use this attention score directly as a logit for predicting $t_3$, but since L12H12

is not explicitly optimized for this task, we instead train a bilinear probe (Hewitt and Liang, 2019) with an analogous structure from scratch.

Concretely, our probe predicts the 3rd move in two steps:

1. **Predicting the target square:** Given the target square of the 1st move $t_1$, which we can extract from Leela's policy output, we predict the 3rd move target square $t_3$ using

$$\Pr(t_3 = y | t_1) = \operatorname*{softmax}_y \left( (h_y^L)^T U^T V h_{t_1}^L + c \right) , \tag{1}$$

where $U$, $V$ are learned matrices with shapes matching $W_Q$ and $W_K$, and $c$ is a learned bias. This is exactly analogous to how the attention weight between $t_3$ and $t_1$ is computed (except for the bias).

2. **Predicting the source square:** We predict the 3rd move source square $s_3$ conditioned on the predicted target square $t_3$ using an analogous bilinear form with separate weights $U'$, $V'$, $c'$:

$$\Pr(s_3 = y | t_3) = \operatorname*{softmax}_y \left( (h_y^L)^T U'^T V' h_{t_3}^L + c' \right) . \tag{2}$$

We train the two probes separately, so the second probe uses ground truth values for $t_3$ during training. At test time, we first predict $t_3$ and then use that to predict $s_3$. See Appendix F for details on training hyperparameters.

Figure 8 shows the accuracy of our probe for predicting the 3rd move target square after each layer of the network. Accuracy mostly increases through the layers, peaking at $(92 \pm 1)\%$ after layer 12. As a simple baseline, probes trained on a randomly initialized copy of Leela achieve only $(15 \pm 2)\%$ accuracy. This shows that the performance can't just be due to the probe "doing all the work."

## 4 Related Work

**Chess-playing neural networks** Our work relies on the Leela networks (Leela Chess Zero team; Monroe and Chalmers, 2024). Leela is based on earlier work on Alpha Zero (Silver et al., 2018) but is much stronger. Ruoss et al. (2024) recently trained a neural network to play chess without any *external* search (such as MCTS). Their network is similar in architecture and strength (lepned, 2024) to the version of Leela we study, so our findings suggest the possibility that their network may have learned to perform *internal* look-ahead or search. Several works have trained networks to play chess using an autoregressive approach motivated by language modelling (Noever et al., 2020; Toshniwal et al., 2021; Feng et al., 2023; Stöckl, 2021; Karvonen, 2024). Unlike Leela or Alpha Zero, these networks don't get the current board state as an input, only a sequence of moves, which makes their task more difficult.

**Learned look-ahead and search** Pal et al. (2023) showed that future tokens are to some extent decodable from hidden representations of a language model at earlier token positions, a finding that relates to our probing results in Section 3.3. However, they don't focus on whether representations of future tokens causally influence the prediction of the current token, which is the core focus of our study on look-ahead. Brinkmann et al. (2024) find an interpretable reasoning algorithm in a small transformer trained to find paths in trees. This suggests similar algorithms could be present in Leela as well, but Leela and chess-playing are much more complex than this model and synthetic task, and so giving as detailed an interpretation as Brinkmann et al. do would be much more challenging. In concurrent work to ours, Taufeeque et al. (2024) and Bush et al. (2024) demonstrate look-ahead in Sokoban-playing RNNs using linear probes. Finally, there is a long line of work showing that neural networks can learn to implement learning algorithms in context, e.g., for regression (Hochreiter et al., 2001; Akyürek et al., 2023; von Oswald et al., 2023b,a) and reinforcement learning (Duan et al., 2016; Wang et al., 2016; Lee et al., 2023). This is algorithmically quite distinct from look-ahead as we study it, and many of these works focus on behavioral evaluations rather than mechanistic analysis.

**(Mechanistic) Interpretability** Methodologically, our work relies heavily on mechanistic interpretability tools, particularly activation patching (Vig et al., 2020; Geiger et al., 2021; Meng et al., 2022). In terms of results, there have also been a few relevant works; most closely connected is

Brinkmann et al. (2024) as already discussed. Examples of interpretability applied to game-playing models include McGrath et al. (2022) and Schut et al. (2023), who study the internals of AlphaZero. But their specific experiments are very different from ours and they do not specifically analyze the potential for search or look-ahead. Recent work has also found that game-playing models trained on move sequences can learn to keep track of the current board state in Othello (Li et al., 2023; Nanda et al., 2023) and chess (Karvonen, 2024). This is orthogonal to our work: Leela already gets the current state as its input, rather than a sequence of moves, and instead of board state tracking, we find evidence of look-ahead to *future* moves.

## 5  Conclusion

We have shown correlational and causal evidence of learned look-ahead in Leela. While we do not have a good understanding of the exact algorithms Leela has learned, our three lines of evidence strongly suggest that in many tactically complex states, some form of look-ahead plays an important role in determining Leela's policy. Some of the techniques we use are very general and might be useful for mechanistically studying complex behaviors in other networks.

**Limitations**   In our mind, the main limitations of our work are as follows: (1) We do not present a precise description of how look-ahead might be implemented in Leela. Understanding this would be interesting from an interpretability perspective and also provide additional evidence. (2) We focus on look-ahead along a single line of play; we do not test whether Leela *compares* multiple different lines of play (what one might call *search*). (3) We focus on board states that are unusually complex and thus more likely to require look-ahead. To understand the role of look-ahead across the entire input distribution, we would need to gain a better understanding of how look-ahead might be combined with simple heuristics in Leela. (4) Chess as a domain might favor look-ahead to an unusually strong extent. It would be interesting to study whether language models use similarly principled mechanisms when appropriate. We think all of these questions could be fruitful directions for future work.

**Impact**   We expect our results to inform future research and discussion rather than having direct societal impacts. There has been significant debate about the degree to which frontier neural models, such as large language models (LLMs), internally implement principled algorithms. Our results on a chess-playing model certainly don't allow immediate conclusions about LLMs, but they are evidence of complex algorithmic mechanisms in neural networks "in the wild," i.e., not trained specifically to demonstrate such mechanisms. Learned optimization (or *mesa-optimization*) could also pose novel risks (Hubinger et al., 2019). Leela or similar networks might be promising candidates for test beds to study such potential risks in toy settings.

## Author contributions

Erik led the project, including developing ideas, executing experiments, and writing. Shreyas built the infrastructure necessary to run interpretability experiments on Leela and contributed ideas and technical support throughout the project, in particular using a weaker model to filter puzzles and find corruptions. Vasil ran several exploratory experiments, first noticed L12H12, and conjectured that it was moving information from 3rd to 1st move target. Cam and Scott gave regular feedback and input throughout the project and helped develop the story for the paper; Scott also helped implement the puzzle filtering and L12H12 ablations. Stuart advised the project. Stuart, Scott, Shreyas, and Erik were all involved in the original conception of the project.

## Acknowledgments and Disclosure of Funding

We thank Michael Cohen, Lawrence Chan, Erik Jones, Alex Mallen, Neel Nanda, and everyone else we talked to for feedback at various stages of this project. We also thank Discord user masterkni6 for providing information to help us finetune our version of Leela (Appendix A). This work was supported by funding from Open Philanthropy, the Future of Life Institute, and the AI2050 program at Schmidt Futures (Grant G-22-63471).

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

# A    Details of the Leela architecture

The version of Leela we use is `T82-768x15x24h-swa-5230000`, which was the strongest officially supported model when this project began. (By now, the newer BT3 and BT4 models are even stronger.) Other versions of Leela are CNNs similar to AlphaZero, but this model is transformer-based, and we will focus on its architecture. For some training details on a newer iteration of Leela, see Monroe and Chalmers (2024), though note that our version has slight architectural differences to that newer one.

The model is available from `https://lczero.org/play/networks/bestnets/`. The network weights themselves don't have a specified license to the best of our knowledge; the code base is under the GPL-3 license, and the training data is under the Open Database License.

Note that this version of Leela wasn't directly trained using MCTS; instead, it was trained using supervised learning on MCTS roll-outs produced by an earlier model. This is a similar objective, however, given that during MCTS, the value and policy nets are also trained to match the output of the entire search process.

**High-level description**    Leela takes a board state as input and produces a distribution over moves and a value (win/draw/loss probability) for that state. It has a policy and a value head, which share a transformer as the main body of the network. Each square of the chess board is represented as a sequence position in this transformer.

**Input encoding**    To feed a board state into Leela, it is turned into 12 bitmaps, one for each piece type of either color. For example, there is a bitmap for white pawns, which is an $8 \times 8$ boolean array with a 1 on each square with a white pawn and a 0 otherwise. In addition to these 12 bitmaps, there are a few channels for encoding castling rights and similar information (these channels have a constant value across all 64 squares). The original version of Leela takes in past board state as well, but we finetuned it to do without those, as we'll discuss. The positional encoding of the model we use is domain-specific. Each square has a 64-dimensional embedding with binary values that encode which other squares could be reached from the given square in one move by some type of piece. The input and positional encodings are concatenated and then fed into a linear map into the 768-dimensional residual stream. Thus, the precise choice of input and positional encoding likely does not matter much; the model can learn how to best embed this information. Note that piece colors are encoded as "player color" and "opponent color", rather than white and black. Additionally, the board is rotated so that the current player's side is always at the same indices. Thus, Leela doesn't need to learn the symmetry between white and black; this is already built into the input encoding.

**Main network**    The main part of Leela is a 15-layer transformer with a residual dimension of 768, 24 attention heads per layer, and a head dimension of 32. Notably, the MLPs have a hidden dimension of only 1024, rather than the common $4 \times d_{\text{residual}}$. Leela is thus unusually heavy on attention heads. The LayerNorms in Leela are applied directly to the residual stream rather than to layer inputs. This is what the original transformer paper did (Vaswani et al., 2017) but unlike most modern transformers. These non-linearities on the residual stream make certain interpretability techniques more difficult to apply but don't present an obstacle for any of our experiments.

**Smolgen**    The biggest way in which Leela differs from a typical transformer is a method unique to Leela called *smolgen*. Like other transformers, Leela produces attention scores using $Q$ and $K$ matrices. However, before applying the softmax, smolgen adds the output of an MLP to these attention scores. This MLP takes in the entire residual state, which is meant to make it easier for the network to use global information when computing attention scores. From an interpretability perspective, smolgen would likely make it much more difficult to understand how attention patterns are computed. However, we don't study that question and can mostly ignore smolgen (i.e. we simply use the combined attention pattern without worrying about where it came from). In the case of L12H12, we used the observed attention pattern to motivate our bilinear probe architecture. It turns out that the part of the attention pattern produced by the (bilinear) query/key mechanism is indeed enough to read off the 3rd move target square.

**Policy head**    The policy head of Leela consists of two 2-layer MLPs, which we'll call the *source MLP* and the *target MLP*. They share the first linear layer for parameter efficiency but have separate second layers. Both layers have output dimension 768 (matching the residual stream). These MLPs

are applied separately to each square, yielding outputs with shape $64 \times 768$ for each. These "source" and "target" outputs are then matrix multiplied along the 768-dimensional axis to yield a $64 \times 64$ dimensional output. Each entry in this output represents the logit for the move from the corresponding source square to the target square. Finally, logits for illegal moves are masked out by setting them to negative infinity, then a softmax over the logits yields the distribution over moves. There are a few additional pieces of machinery to deal with pawn promotion, but these aren't particularly important for our purposes.

**Value head** We don't use the value head in the main paper but briefly discuss it here for completeness and will present results in Appendix C. It begins by projecting the 768-dimensional residual stream down to 32 dimensions, using a learned linear map that's shared across squares and applied separately to each square. The 64 32-dimensional embeddings are then rehaped to get a single 2048-dimensional vector. This vector is passed through a small MLP, which produces three logits, for the probability of the current player winning, drawing, and losing.

**Finetuning to avoid reliance on past board states** Leela originally takes in the past 8 board states, rather than only the current one. This mostly shouldn't be necessary to play chess well, though it might help a bit given the limited computational capacity. For example, knowing the opponent's last move may make it easier to tell what threats (if any) they currently have. For our purposes, passing in a history of past board states is very inconvenient since we want to automatically generate corrupted states for activation patching. Generating corrupted histories instead would be much more challenging. We experimented with different options, such as simply passing in zeros for the history, repeating the current board state, or synthesizing a valid (but not semantically meaningful) history. We found that in most cases, Leela's output didn't depend significantly on what we passed in as a history, indicating that Leela (perhaps unsurprisingly) wasn't making too much use of past board states. However, in some cases, the output changed a lot when we modified the history, for unclear reasons.

To avoid such confounders when activation patching, we finetuned Leela to behave the same whether or not any history was passed in. Then, for all our experiments, we used this finetuned model and didn't pass in any history. The finetuned model matches the original one in strength despite not using history (both in terms of solving puzzles from our dataset, as well as when playing against the original model directly). Anecdotally, it also seems that finetuning didn't change the model's mechanism too much (e.g., some attention heads that we had interpreted before finetuning still seemed to perform the same function afterward).

## B  Leela's playing strength

Leela is usually evaluated with MCTS, in which case it is the strongest MCTS-based chess engine, and competes with Stockfish—a classical chess engine making heavier use of external search—for the title of strongest chess engine in general (Haworth and Hernandez, 2021).

There are fewer evaluations of the strength of only the neural network itself. A Lichess bot using only Leela's policy network has achieved blitz and rapid ratings over 2600, see `https://web.archive.org/web/20240519065001/https://lichess.org/@/LazyBot/all`. This is clearly below the strength of the best human players but far better than most amateurs. The version we use is slightly larger than the one underlying the Lichess bot and likely somewhat stronger.

Ruoss et al. (2024) recently trained a neural network that can also play chess extremely well without external search. Lepned (2024) compared this network to Leela on the task of solving tactics puzzles and found that the strongest Leela networks outperformed the network by Ruoss et al. (2024), but the version we use (T2) seems roughly evenly matched.

## C  Value head results

As discussed in Appendix A, Leela has a policy head and a value head on top of a shared network body. For simplicity, we only used the policy head in the main paper: the metric we measured there were always the log odds that the policy head assigns to the correct move. But of course, most of the computation going into the network's policy output happens in the shared body (since the policy

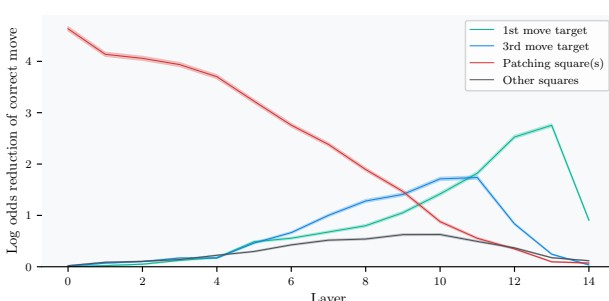
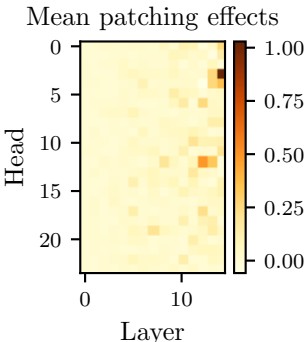

Figure 9: Residual stream patching results (analogous to Fig. 3 but using the win log odds instead).

Figure 10: Attention head patching results (analogous to Fig. 4 but using the win log odds instead).

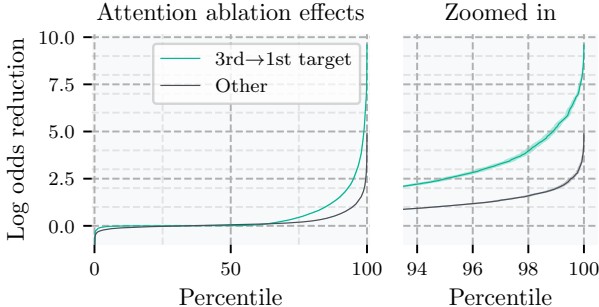

Figure 11: L12H12 ablation results (analogous to Fig. 5 but using the win log odds instead).

head itself is very small), so we'd expect those computations to also affect the value head. We should thus expect overall similar results when using the value head to measure effects of interventions. In this appendix, we show that this is indeed the case.

Our metric in this appendix are the log odds of the win probability given by the value head, i.e., $\log\left(\frac{p_{\text{win}}}{1-p_{\text{win}}}\right)$, where $p_{\text{win}}$ is computed by applying a softmax to the three output logits of the value head (for win/draw/loss).

Figures 9 to 12 show all the results from the main paper using this win probability metric instead of the probability of the correct move. In other words, the interpretation of these figures is the same as in the main paper, except that "log odds reduction" refers to the win probability log odds.

The only new finding is that L14H3 seems very important to the value head (Fig. 10). But note that L12H12 is still just as important as it was for the policy head (log odds reduction of 0.49 when we ablate it), it's only *relatively* less visible in the heatmap.

The probing results in Section 3.3 only use the shared network body rather than either of the heads, so there is no separate value head version of them.

# D    Creating the puzzle dataset

As described in Section 2.2, we are looking for states where Leela finds the correct move but a smaller model does not. The small model we use is "Little Demon 2", a CNN with only 390k parameters. Here, we describe the exact procedure we used to create this dataset.

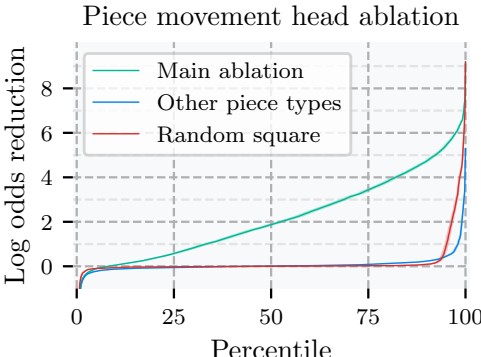

Figure 12: Piece movement head ablation results (analogous to Fig. 7 but using the win log odds instead).

We begin with a dataset of Lichess tactics puzzles(Lichess team), available at `https://database.lichess.org/#puzzles` under a Creative Commons CC0 license. These puzzles have been automatically generated from human games using a variety of heuristics and computer analysis to select "interesting" states. The *player color* (i.e. the color whose turn it is in the starting state) always has an advantage, but only one move maintains that advantage.

Many of these puzzles are still easy to solve using heuristics. That is where the smaller version of Leela comes in. We discard any puzzle where this smaller version assigns more than 10% probability to any of the player color's moves from the principal variation. In other words, we require *every* move the player needs to make to be "difficult to find". Note that we do *not* apply this restriction to responses by the opponent. Typically, the principal variation has two moves by the player, sometimes three or rarely more.

An alternative would have been to use puzzle ratings that Lichess provides to judge difficulty—these are based on many human players attempting to solve puzzles. An advantage of our approach is that it is more directly related to what's difficult for neural networks and avoids potential human idiosyncrasies. It is also a much more general method that's still applicable if human difficulty ratings are not available.

In addition to making puzzles "difficult", we also want to ensure that the large version of Leela gets them right. Otherwise, there is no interesting behavior for us to study with interpretability. We thus discard any puzzles where Leela assigns less than 50% probability to any of the player's moves. In particular, Leela's top choice always coincides with the best move in our dataset.

Finally, for some of our experiments it is useful if the principal variation is *forcing*, in the sense that there is only ever one clearly best move at each step. In particular, otherwise it would be fundamentally impossible to predict the 3rd move well since it would depend on the opponent's response. We thus also discard puzzles where the small model assigns less than 50% probability to the 2nd move in the principal variation, i.e. the opponent's response. For the player's moves, the Lichess dataset already guarantees that there is only one winning move at each step.

This filtering procedure and the various thresholds were chosen based on manual inspection of a few puzzles; we tried to ensure that the puzzles seemed intuitively "interesting" to us. We did *not* tune this procedure based on any interpretability results.

# E   Automatically generating corrupted states

**Generating candidate corruptions**   Given a board state, we generate candidates for corrupted states by applying each of the following mutations separately (i.e. each mutation yields one candidate corruption; each candidate has only one mutation applied):

1. Add a single pawn of either color on an empty square.

2. Remove any pawn from the board.
3. Move any non-pawn piece to any other empty square.

We discard candidates that lead to illegal states (e.g. if the opponent of the current player is in check). This produces a few hundred corruption candidates in a typical board state. The reason we consider only these corruptions and not, for example, moving pawns or adding non-pawn pieces is simply to keep the search space manageable and because we almost always find a "good" corruption using only these candidates.

**Filtering candidates**   We apply several corruption filtering steps using the full version of Leela as well as the weaker model we used for filtering puzzles. As with puzzle filtering, we manually designed these filters to lead to intuitively "interesting" corruptions, but did not tune them in any way on interpretability results. The precise numerical cutoffs used in the filters are somewhat arbitrary.

1. We keep only corruptions that reduce the probability Leela assigns to the previously best move to less than 10%—if a corruption doesn't make the previously best move bad, there's no reason to expect activation patching to have any effect.
2. We discard corrupted states where the weak model's log odds of the previously best move decrease by more than 0.2. Anecdotally, these corruptions often make the best move worse for "obvious" reasons, such as placing an opponent pawn that directly attacks the target square of the move.
3. We discard corruptions that make the board state significantly *better* according to Leela's value output (increasing the difference between win and loss probability by more than 0.1). These corruptions often make the previously best move worse simply by making some other move extremely good, e.g., putting the opponent's queen on a square where it can be captured.

**Picking the "lowest impact" corruption**   Out of all the remaining corruptions, we pick the one such that the weak model's move distribution changes as little as possible since we are looking for corruptions that change the best move for "subtle" reasons that the weaker model doesn't "notice". Concretely, we minimize the Jensen-Shannon divergence between the weak move distribution on the clean and corrupted state.

## F   Probe training details

We train one probe for each layer, on the residual stream activations after the MLP of that layer. More precisely, we use the activations after the LayerNorm (recall from Appendix A that Leela applies a LayerNorm to the entire residual stream, rather than to module inputs). We use 70% of our puzzle dataset for training and the remaining 30% to evaluate probes.

We found that the choice of hyperparameters has essentially no effect on probe accuracy; we used Adam with a learning rate of 1e-2, no weight decay or other regularization, a batch size of 64, and trained for 5 epochs.

The probe can be thought of as a low-rank bilinear form, where we parameterize the low-rank matrix as $U^T V$ for $U, V$ matrices of shape $k \times d$, with $d = 768$ the residual stream dimension. We used $k = 32$ for the rank, simply to match the attention head dimensions of Leela (each head's attention pattern can analogously be interpreted as the result of a low-rank bilinear form). Similarly to other hyperparameters, we found that other values of $k$ work about equally well.

## G   Details on confidence intervals

All errors we report are $2\sigma$ or 95% confidence intervals, depending on context. We have three types of error:

1. For the error of an average, such as the average effects of residual stream activation patching (Fig. 3), we report two times the standard error of the mean. Given that these are means over thousands of data points from i.i.d. samples, this $2\sigma$ error likely corresponds well to a $\sim$95% confidence interval (though we don't report it as such).

2. For percentile plots that show a distribution (Figs. 5 and 7), we report an approximate 95% confidence interval based on the fact that the number of data points below a given percentile is binomially distributed. We will describe the exact procedure for this shortly.

3. For our probes, we consider the error from randomness during probe initialization and training, in addition to the error when estimating the test accuracy from finitely many samples. We compute the standard error of the mean for both and propagate them (i.e. add the square errors) to get an overall error; we then again report twice that error to approximate a 95% confidence interval. See details below.

Note that all of these errors are statistical, and due to our sizable puzzle dataset (22.5k puzzles), they are mostly very small. Naturally, there might be unknown systematic errors that dwarf these statistical ones.

**Error bars for probe accuracies** We consider two sources of error: (1) randomness during probe initialization and training, and (2) estimating the accuracy of the final trained probe from our finite evaluation dataset. For (1), we have five training runs and compute a standard error of the mean over these. For (2), we compute the standard error as well (noting that the empirical accuracy is an estimate of the mean of a Bernoulli random variable using the samples from the evaluation dataset). Then we compute a total error using error propagation as $\sigma_{\text{total}} = \sqrt{\sigma_{\text{train}}^2 + \sigma_{\text{accuracy}}^2}$. For error bars, we report a $2\sigma_{\text{total}}$ interval. $\sigma_{\text{train}}$ dominates especially for the randomly initialized model; we could reduce the error bars further by training more than five probes, but they are already negligible compared to the effect size.

**Confidence intervals for percentiles** Several of our results visualize a distribution by plotting a percentile function (the inverse of a cumulative distribution function), see Figs. 5 and 7. We approximate a 95% confidence interval for each percentile. Let $n$ be the number of samples we use; in all our cases, these are i.i.d. from an unknown distribution (whose percentile function we want to plot). The number of samples $k(p)$ below a given percentile $p \in [0, 1]$ follows a binomial distribution $B(k(p); n, p)$. We find the 2.5%-th percentile and 97.5%-th percentile of this binomial distribution—$k(p)$ lies between these values with 95% probability. We then sort all our samples by value and find the samples at the 2.5%-th and 97.5%-th percentile by value. Those values form the lower and upper bound for our confidence interval of the $p$-th percentile. This is an approximation since the samples whose values we use will never be *exactly* at the 2.5%-th and 97.5%-th percentile, but we have at least several thousand samples in all cases, so the approximation should be very close to a 95% interval.

# H    Results on subsplits of the puzzle distribution

As discussed in the main text, we noticed that the results of all our experiments are noticeably different on puzzles where the 1st and 2nd move target square are different. In 83% of our overall dataset, these squares coincide because many of these puzzles contain sacrifices by the starting player.

In this appendix, we show results on both subsplits of the data, the one with the same 1st and 2nd move target, and the one with different 1st and 2nd move targets. We find that the effects we observe are generally much stronger in cases where 1st and 2nd target square are the same.

In most cases, the effects are qualitatively similar on both subsplits. The one exception is L12H12, where the baseline ablation has a stronger effect than the one between 3rd move target and 1st move target (though keep in mind that the baseline is ablating 4095 weights, vs only one weight for the main ablation). In fact, the baseline has bigger effects in this setting with different 1st and 2nd target squares than on the full dataset, unlike all other ablation effects we see. This might suggest that L12H12 is performing different functions in these cases (though that function seems less important, given the much lower effect size when activation patching L12H12 in the "different targets" setting).

One natural question is whether any of the results we found in the main text involving the 1st move target square instead apply to the 2nd move target square when they are different. This does not seem to be the case to a meaningful extent. In the residual stream patching results, the 2nd move target square is automatically included in the "other squares" baseline on the "different targets" split, but unsurprisingly, the 1st move target effects are still much larger than this baseline. (Logits are read off

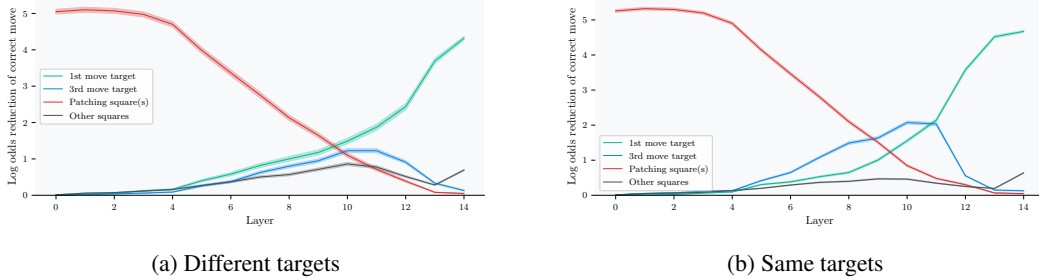

(a) Different targets            (b) Same targets

Figure 13: Residual stream patching results, analogous to Fig. 3.

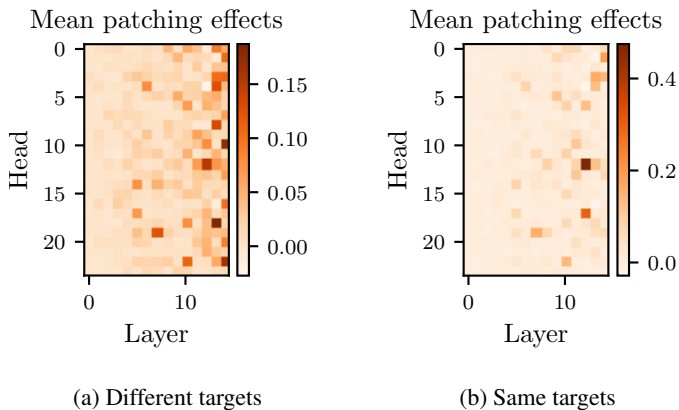

(a) Different targets            (b) Same targets

Figure 14: Attention head patching results, analogous to Fig. 4.

on 1st move target squares after all.) We also tested ablating information flow in L12H12 from the 3rd move target to the 2nd move target, but the effect on the "different targets" split is even smaller than that of ablating information flow to the 1st move target. It thus seems likely that the first, rather than the 2nd target square, is critical for the results presented in the main text, even though they often overlap there.

# I  Software libraries used

We use `onnx2torch` (ENOT developers et al., 2021) to convert the official Leela ONNX models to PyTorch (Ansel et al., 2024) for easier instrumentation. This produces auto-generated code for the PyTorch forward pass, which we manually adjust in a few ways to support our interpretability experiments. We then use `nnsight` (Fiotto-Kaufman) to implement our interventions. We also build on the `lczero_tools` package (Graffa) for some of our instrumentation. We use `python-chess` (Fiekas)

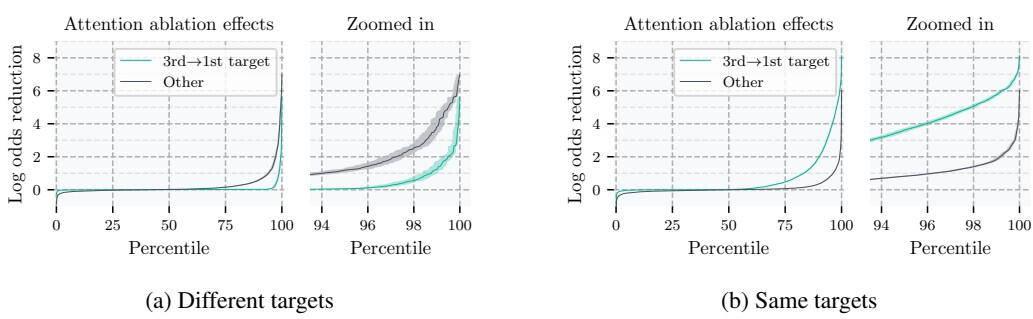

(a) Different targets            (b) Same targets

Figure 15: Ablations in L12H12, analogous to Fig. 5.

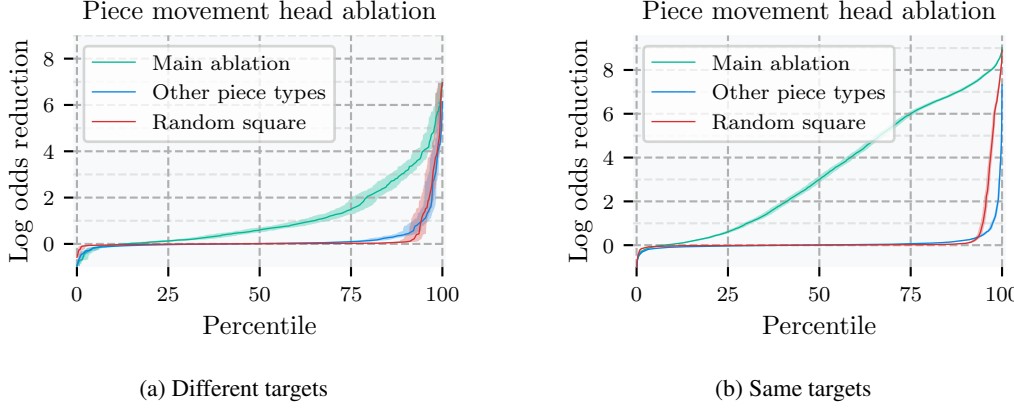

(a) Different targets

(b) Same targets

Figure 16: Ablations in piece movement heads, analogous to Fig. 7.

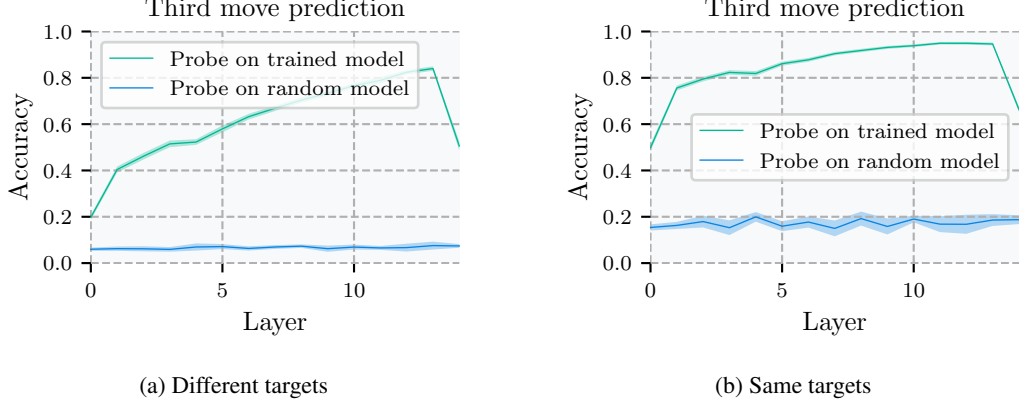

(a) Different targets

(b) Same targets

Figure 17: Probing results, analogous to Fig. 8.

for chess logic, and `einops` (Rogozhnikov, 2022) for conveniently implementing some of our methods. The figures are produced using `iceberg` (IceBerg Contributors, 2023) and `matplotlib` (Hunter, 2007). See `https://github.com/HumanCompatibleAI/leela-interp` for the full list of packages we use.

