# OpenReview forum: "Evidence of Learned Look-Ahead in a Chess-Playing Neural Network"
_NeurIPS.cc/2024/Conference — NeurIPS 2024 poster_

### Official Review · Reviewer_p7Jw · 2024-07-13

**Soundness:** 2
**Presentation:** 3
**Contribution:** 3
**Rating:** 4
**Confidence:** 4

**Summary:**

The author examine if a chess NNs encode future moves in their activations. The results suggest that Leela predicts future self-play moves and that the activations can be manipulated to change the predicted moves.

**Strengths:**

# originality

Linear probes for difference concepts are well known in the chess literature, but the activation patching and explicitly looking at future moves as concepts are novel.

# quality

The analysis looks good, but the lack of consideration of alternative hypothesizes or attempts to disprove the results weakens the results.

# clarity

The paper is clear and flows well.

# significance

I think this work is interesting and presents some good ideas for chess AI and the broader XAI community.

**Weaknesses:**

My main concerns with this paper is that it does not test the methods outside of a very limited scope. Even looking a puzzles where Leela gets the next move wrong would be a good first step, i.e. what is the accuracy of your move predictor when the model is wrong, or can you patch to get the correct move.

I'm also not that surprised by this result and don't think the main motivation is addressed. The Leela NN is trained to predict the future actions of itself, and those are based on an explicit tree model of the game MCTS. The model internally doing a depth 3 search seems very plausible and I don't think proves the motivating claim of "look-ahead in neural networks", as the heuristics could simply be on the depth 3 tree instead of the depth 1 tree. One options for this analysis would be to look at a chess NN that's not trained on selfplay data (Maia, CrazyAra, etc).

**Questions:**

Could you give a more formal definition of " look-ahead in neural networks" and explain how this work proves it?

Does the move probe work on human games or non-puzzle positions?

Figure 8, what is the accuracy measuring? There are 64 squares on a chess board, how is random getting above 2% accuracy?

What would a falsification look like in this method? The patching for example seems so show that you can disrupt the chosen square, but this assumes the activations are encoding this linearly. If they were non-linear or only partially in the patch how would that changes these results?

**Limitations:**

Discussed above

---

> ### Author Rebuttal · Authors · 2024-08-06
>
> Thank you for your review and feedback!
>
> > looking a puzzles where Leela gets the next move wrong would be a good first step, i.e. what is the accuracy of your move predictor when the model is wrong, or can you patch to get the correct move.
>
> We focus on puzzles that Leela gets right simply so we can apply our interpretability methods. For example for the move predictor probes: what should be the “ground truth” for the 3rd move in cases where Leela gets the puzzle wrong? We could see if the probe can still predict the 3rd move of the correct continuation, but given that Leela got the puzzle wrong, there’s no reason to expect this. Leela might still use look-ahead along some incorrect continuation, so we could ask how good the probe is at predicting the 3rd move in this incorrect line. But of course, there are many different incorrect lines (vs one correct line), and we don’t have ground truth for which, if any, Leela is considering. For patching, it will massively depend on how invasive an intervention we perform. For example, patching the right squares in the final layer could just override the move output to be whatever we want.
>
> >  The model internally doing a depth 3 search seems very plausible and I don't think proves the motivating claim of "look-ahead in neural networks"
>
> We don’t follow; if the network is doing a depth 3 search (over possible move sequences), we’d certainly consider that an instance of look-ahead in neural networks. Note that we are not claiming look-ahead arbitrarily many steps into the future, is that the point of confusion?
>
> > Could you give a more formal definition of " look-ahead in neural networks" and explain how this work proves it?
>
> By look-ahead in neural networks, we mean internally representing future moves (of the optimal line of play) and using those representations to decide on the current move (see line 45). Our probing results (section 3.3) are our most straightforward evidence for representations of future moves, though the residual stream patching experiments (section 3.1) also suggest this already. The patching experiments in section 3.2 give evidence of such representations being used the way we’d expect, and in particular to decide on the current move. For example, L12H12 seems to move information from future move target squares back to the target square of the next move, and a very targeted ablation in this head has outsized effects on the output.
>
> > Does the move probe work on human games or non-puzzle positions?
>
> All puzzles in the Lichess dataset we use come from human games, they are simply selected to be tactically interesting states. In many chessboard states, there are many plausible lines of play, so predicting several moves into the future is fundamentally much less feasible. Using puzzles means that at least there is a clear ground truth correct line, and we can then check whether Leela represents it.
>
> > Figure 8, what is the accuracy measuring? There are 64 squares on a chess board, how is random getting above 2% accuracy?
>
> The “probe on random model” line in Fig. 8 is a probe trained on a randomly initialized model, which is different from a randomly initialized probe (or random predictions). Even the activations of a randomly initialized model contain information about the input, they just don’t contain more sophisticated features. So this baseline checks how well future moves can be predicted just by the probe itself, without interesting features from Leela.
>
> > What would a falsification look like in this method?
>
> If the experiments we ran didn’t yield non-trivial effects, we’d consider that evidence against look-ahead. For example, a probe trained to predict future moves could have achieved accuracy not much better than the baseline probe, or patching on future move squares could have had effects no bigger than patching on other relevant squares.
>
> > the lack of consideration of alternative hypothesizes or attempts to disprove the results weakens the results.
>
> We did implicitly consider alternative hypotheses in every one of our experiments. For example, for the residual stream patching experiment (section 3.1), we wondered whether future move target squares only have big patching effects because these squares tend to be “obviously”/”heuristically” important in the puzzle starting state. That is why we used a very strong baseline, by taking the maximum effect over other squares. While future move target squares might often be important for heuristics, so are many other squares, and under a non-look-ahead hypothesis, it seems likely that the maximum over other squares should typically be larger than the importance of one specific future move square. We have similar baselines to rule out simple alternatives in all our experiments.
>
> We realize that this reasoning might not always be apparent in the paper, so we’ll include the motivation for baselines more explicitly. Thanks for drawing attention to this!
>
> Of course, we can never fully prove that there's no alternative explanation for our results, we can only rule out specific alternative hypotheses. We don’t know of any concrete hypotheses not involving look-ahead that predict or explain our results, but we would be very curious in case you have a specific one in mind.
>
> > The patching for example seems so show that you can disrupt the chosen square, but this assumes the activations are encoding this linearly. If they were non-linear or only partially in the patch how would that changes these results?
>
> There seems to be a misunderstanding here. We do not assume any form of linearity for our patching results. We do operate under the hypothesis that information about future moves is localized to their squares, but this is something we argue for with our results rather than an assumption (see line 128).

---

> > ### Comment · Reviewer_p7Jw · 2024-08-09
> >
> > Thank you for your detailed response. I continue to be concerned that this work is not falsifiable as presented. You only look at features in a highly specific dataset (Lichess puzzles that the model is successful on) and never checked to see if the features are absent when not looking at the dataset. This is also a result of your definition which considers only the positive case, i.e., a model that represents all lines equally before discarding the results and deciding independently would also meet your definition. This is the _"Leela might still use look-ahead along some incorrect continuation"_ scenario that you mentioned.
> >
> > In particular I am concerned by the lack of "real' games, the Lichess puzzles dataset only contains positions that meet certain easy to evaluate heuristics. It is impossible to determine if you are observing these heuristics or more robust features in this test. I would be much more confident of a claim of general look-ahead if they showed the effects outside of a cherry-picked set of samples. Explicitly, presenting a hypothesis for this work and explaining how your tests would disprove it would greatly improve things on this front, currently a negative result simply suggests you're not searching hard enough.

---

> > > ### Author Response · Authors · 2024-08-09
> > >
> > > Thank you for the follow-up!
> > >
> > > > I continue to be concerned that this work is not falsifiable as presented. You only look at features in a highly specific dataset (Lichess puzzles that the model is successful on) and never checked to see if the features are absent when not looking at the dataset.
> > >
> > > Is your concern (1) that we should check that we get *negative* results on inputs where there is no look-ahead, as a sanity check/baseline? Or (2) that we should present *positive* results on a wider distribution?
> > >
> > > If (1), it's unfortunately unclear how to find inputs where we can rule out that look-ahead is happening. But we do have other types of baselines in each experiment.
> > > If (2), as described above, we only claim look-ahead on this specific distribution. So in this case, the question in our mind would be how interesting this claim is, rather than issues with our evidence.
> > >
> > > Please let us know in case we misunderstood your point! We unfortunately still don't see why you think our work "isn't falsifiable."
> > >
> > > > a model that represents all lines equally before discarding the results and deciding independently would also meet your definition. This is the "Leela might still use look-ahead along some incorrect continuation" scenario that you mentioned.
> > >
> > > We don't think so. If a model discarded results from considering future lines, then the representations related to those future lines would not influence the output of the model. So we would not consider this look-ahead under our definition (because it's lacking the "using those representations to decide on the current move" part). We perform interventions to check this in sections 3.1 and 3.2, so for such a model, we also wouldn't get most of the positive evidence of look-ahead we present.
> > >
> > > The "Leela might still use look-ahead along some incorrect continuation" scenario is different: here, Leela would not be *discarding* the look-ahead along incorrect lines, but instead would be using it to output an (incorrect) move. Please let us know in case we misunderstood your concern.
> > >
> > > > the Lichess puzzles dataset only contains positions that meet certain easy to evaluate heuristics. It is impossible to determine if you are observing these heuristics or more robust features in this test
> > >
> > > Could you clarify what you mean by "easy to evaluate heuristics?" In case you mean chess heuristics to easily find the best move (without look-ahead), then (1) as we've described in the paper (section 2.2), we've specifically tried to make the puzzles *not* solvable by simple heuristics, and expect they are much more difficult than typical chess states, and (2) our experiments give evidence of look-ahead without assuming that look-ahead is required to solve those puzzles (see also the beginning of our response to SpQ5).
> > >
> > > > Explicitly, presenting a hypothesis for this work and explaining how your tests would disprove it would greatly improve things on this front, currently a negative result simply suggests you're not searching hard enough.
> > >
> > > Could you say more about what kind of hypothesis you are looking for here? A complete hypothesis in the sense of pseudocode for how Leela might play chess this well without look-ahead is clearly infeasible: all known algorithms for this level of chess performance involve explicit look-ahead/search, so if Leela wasn't using look-ahead, it would have to use some method or set of heuristics that haven't been discovered in decades of chess engine development. (This is certainly plausible a priori, but we naturally couldn't specify any concrete such hypothesis.)
> > >
> > > That is why we've focused on hypotheses to "explain away" our specific results in our response above (see the alternative explanation we considered and falsified for our results in section 3.1). As mentioned, we now intend to include this in the paper to make it explicit. If this kind of hypothesis is not what you are suggesting, it would be very helpful if you could elaborate a bit more. Thank you!

---

> > > > ### Comment · Reviewer_p7Jw · 2024-08-13
> > > >
> > > > Thank you for the response. I have read the other comments and will discuss the paper with the other reviewers. I apologize for not getting back to you sooner.
> > > >
> > > > > Is your concern (1) that we should check that we get negative results on inputs where there is no look-ahead, as a sanity check/baseline? Or (2) that we should present positive results on a wider distribution?
> > > >
> > > > I think either result would strengthen your paper significantly, but I was concerned with the former in my comment. The patching method used for detecting features relies on assumptions about the inner workings of NNs that are based on inferences not formal proofs. So you present good evidence that your methods detect something consistently in the dataset of positive cases, but you never check if they also detect something when the features are absent.
> > > >
> > > > > If a model discarded results from considering future lines, then the representations related to those future lines would not influence the output of the model.
> > > >
> > > > I agree that this is something you test for, but again the test assumes that the influence of squares is linear. So patching one will simply shift the effects by some regular amount. You do not test more complex combinations. I am not asking you to do so, more asking you to check negative cases so that you can better verify the state of your detectors.
> > > >
> > > > > "easy to evaluate heuristics?"
> > > >
> > > > The heuristics are listed here: https://github.com/ornicar/lichess-puzzler/blob/master/tagger/model.py#L6 . They are all simple rules that can be checked quickly, the human games are just used to generate potential positions. Testing more complex, i.e. human curated puzzles would help here.
> > > >
> > > > > kind of hypothesis you are looking for here?
> > > >
> > > > There is previous work showing chess NNs learn heuristics [1], some of which include modeling the future e.g. forks. So some definition of look-ahead that explains how it differs from heuristics and what a model would look like with or without it. Currently you only consider models that have this property.
> > > >
> > > > [1] https://arxiv.org/abs/2111.09259

---

> > > > > ### Author Response · Authors · 2024-08-14
> > > > >
> > > > > Thank you for your follow-up response. We are very grateful for your time, and for the lively discussion about our work!
> > > > >
> > > > > >  but you never check if they also detect something when the features are absent.
> > > > >
> > > > > We apologize if we’re not understanding this request, but we are still not sure what you mean by “when the features are absent”.
> > > > >
> > > > > Our claim is that when (1) we are in a tactically “interesting” position (as defined by our smaller model filtering) and (2) there is a unique principle variation (as evaluated by a powerful engine such as Stockfish) and (3) Leela outputs the correct move, our probe can predict the 3rd move with 92% accuracy.
> > > > >
> > > > > Which one of these three conditions being not present would “make these features absent”? And what would you like to see the probes predict in that case?
> > > > >
> > > > > We appreciate your patience here, and are keen to understand how we can make our result stronger.
> > > > >
> > > > > >  They are all simple rules that can be checked quickly
> > > > >
> > > > > We want to mention here that the board positions from the lichess website are first fed through Stockfish NNUE at 40 meganodes (as mentioned in another response, this computation took 50 years of CPU time by the lichess team). We reference `lichess-puzzler/generator/generator.py` line 100, where an advantage value is calculated from Stockfish’s solution, and line 169 where the real board position is accepted or rejected based on Stockfish’s evaluations, not the tagging. The tagging is done after the fact, and based on Stockfish’s best move.
> > > > >
> > > > > > There is previous work showing chess NNs learn heuristics [1], some of which include modeling the future e.g. forks.
> > > > >
> > > > > From [1], the authors probe for the presence of `pawn_fork`, `knight_fork`, `bishop_fork`, and `rook_fork` (please let us know if we missed a concept).
> > > > >
> > > > > Without loss of generality we will quote the description of `rook_fork` from [1], “True if a rook is attacking two pieces of higher value (queen, or king) and is not pinned.”
> > > > >
> > > > > This is considering the future for one step, from the POV of the current player. We wanted to test if the network considers moves more steps down the line, specifically, if it considers an opponent response, and then the model’s response after, which we believe [1] does not study.
> > > > >
> > > > > Again, we thank you for the rich discussion and follow-up responses, we really appreciate it!

---

### Official Review · Reviewer_ScDd · 2024-07-13

**Soundness:** 4
**Presentation:** 4
**Contribution:** 3
**Rating:** 7
**Confidence:** 4

**Summary:**

**Update after rebuttal:** To me personally, the authors' response clarifies all open questions and misunderstandings and adds interesting new results. I remain convinced that the work is ready for publication and interesting to the NeurIPS audience, which means my score remains 'Accept'.

The paper investigates whether it is possible to reliably identify functional signatures of look-ahead in a Leela Chess Zero (policy) network. This analysis in a complex domain (chess) with a relatively large network is challenging. To do this, the paper uses three interpretability techniques for transformers: activation patching to measure the effect of specific interventions on neuron-activations and the sub-network that they influence, analysis and structured ablation of attention patterns, and training of simple read-out probes to test whether certain information is represented in the internal state or not. Additionally, the paper comes up with a method to automatically collect a large, high-quality dataset of challenging chess situations (that simpler chess engines cannot solve) that have a short and unique solution. This allows reliably identifying candidate-targets to aim the look-ahead investigation at. It also allows automatically constructing highly non-trivial interventions on the board state, which are crucial for meaningful activation patching. The paper finds clear and convincing evidence (from multiple angles and with various control experiments) that at least in these situations, the Leela Chess Zero network performs a one-step look-ahead of its next move (an opponent move is in-between, making it a two-step look-ahead in terms of game steps).

**Strengths:**

* Very well written paper, with clear intro, and explanation of the fairly involved methodology, and great supporting figures.
* Very high-quality case-study of interpretability and analysis of a concrete capability/mechanism “in the wild” (meaning in contrast to networks trained on synthetic data to facilitate or simplify the analysis). By its very nature, interpretability work is typically bespoke, and while some general techniques can be developed, I think the field greatly benefits from a body of well executed case studies that provide at least an abstract recipe and approach for others to adapt.
* Great experimental work, with convincing evidence, important controls, and multiple investigations aimed at the same question to provide strong evidence. Despite the challenging setting, experiments are conducted with care and rigor, and attention to ruling out some obvious alternatives that could lead to the same findings.
* The filtered dataset is a contribution in itself, as well as the particular model used in this paper, which may easily be picked up by follow-up research.

**Weaknesses:**

* By the nature of a case study, the paper’s findings are limited to the particular network and the particular domain. The authors have managed to successfully identify and exploit aspects of the architecture (due to the network architecture, the internal state seems to map well onto a 2D chess board) and the data (filtered puzzle set with unique and short solution trajectories). I think this is crucial for the success of the current study and is highly non-trivial and original work, but naturally this means that the method cannot be straightforwardly applied e.g., to a LLM. I do believe though that the work here is highly inspiring to adapt the techniques to other models and domains.
* (minor) Throughout the paper I was wondering why the policy network and not the value network was used. As the appendix says, both networks share the same torso and simply add relatively shallow heads, meaning the findings in the paper also apply to the value network. I think it would be good to mention this and a few more details in the main paper (such as training via supervised learning, and fine-tuning to remove history-dependence). I will list these below in ‘Improvements’.

**Verdict:**
The paper is a great case-study in interpretability and analysis, and I greatly enjoyed reading it. The question tackled in the paper is challenging, and I personally believe that the paper does a great job at producing a number of convincing pieces of evidence, with well chosen control experiments (such as a random network for the probes, and control-ablations for activation-patching and attention-ablations). While there is a challenge to transfer the method to other settings, the paper helps with this by clearly explaining the reasoning behind the various steps. Ultimately, I appreciate a well executed domain-specific case-study over a sloppy execution in a more general setting. Interpretability research is hard, particularly mechanistic interpretability, and I think that the field will be a composition of general techniques (which are also used in the paper) and exemplary case-studies for others to follow and apply to their research questions and domains. Therefore, I think the current paper may well have significant impact beyond the specific findings, which are limited to a particular network and a particular task. I currently recommend acceptance of the paper: all main claims are critically investigated and supported by evidence, the work is novel and very original, and I think is interesting to a wide part of the NeurIPS audience and may be quite impactful for follow-up work.

**Improvements:**

1. I think it would be nice to mention a few more details about Leela in the main paper. In particular that the torso is the same as for the value network (and the value-head is relatively shallow and has no additional attention mechanism, so findings also hold for the value network). Maybe something like L425-428 in the appendix plus a brief description of the training (supervised on a set of high-quality trajectories, plus fine-tuning to remove history-dependency).
2. It would be interesting to see how the findings regarding look-ahead evolve during training. This is completely optional, and beyond the scope of the paper (and the authors may not have access to the original training procedure / checkpoints). But it would be interesting to see whether attention heads and the activation patching results develop gradually or sharply and whether that can be related to a similar increase in performance on the filtered puzzle dataset used in the paper.

**Questions:**

1. Are there situations where the weaker Leela gets the puzzle right, but the stronger Leela does not? Or is the stronger Leela strictly better?
2. Related: how has the weaker Leela been trained? Is it plausible that, e.g. stronger MCTS during training of the weaker Leela means look-ahead is much less important, hence why it does not develop look-ahead (or does look-ahead only develop at sufficiently large and strong networks)? [It is perfectly fine to respond that you do not have the answers for these questions and not spend any more time on this; I am just curious.]
3. The probe accuracy in Fig 8 is relatively high in early layers - how does this fit with the activation patching results that show the largest effect only in medium-to-late layers 6-11?
4. As far as I am aware the precise details about the training process of this variant of Leela have not been published (in an academic venue). While this is beyond the scope of this paper, having these details (or as many as possible), e.g, in the appendix, would greatly help the scientific community to use this version of Leela as something that can be reliably reproduced. This may help boost Leela’s popularity for *scientific* research using openly available chess models. Similarly, the model and dataset used in this paper (with the fine-tuning) are a nice contribution.
5. L225 typo:  “are seem”.

**Limitations:**

Limitations are nicely discussed in Sec 5.

---

> ### Author Rebuttal · Authors · 2024-08-06
>
> Thank you for your careful review and great suggestions!
>
> >  Throughout the paper I was wondering why the policy network and not the value network was used.
>
> Great question, there was no particular reason for this choice except that we decided to focus on only one head for the purposes of exposition. We have re-run all our experiments with the value head (i.e. using the log odds of the win probability instead of the log odds of the top move as the effect metric). As you suspected, the results are qualitatively the same as for the policy head, see the PDF attachment to the general response. We’ll include these in the paper and also clarify that the network has a shared body and two small heads.
>
> > I think it would be nice to mention a few more details about Leela in the main paper.
>
> Thank you for the suggestion, we agree this would help readers and will move some information from the appendix to the paper!
>
> > It would be interesting to see how the findings regarding look-ahead evolve during training.
>
> We agree this would be very interesting to study, but also that it’s out of scope for this paper. There are public training checkpoints for at least some versions of Leela, so in principle, this would be feasible to study in future work. Applying our exact methods would require finetuning away the history requirement for each checkpoint, which should be possible (albeit somewhat computationally intensive if we want to use many checkpoints in order to even detect potential sudden transitions).
>
> > Are there situations where the weaker Leela gets the puzzle right, but the stronger Leela does not? Or is the stronger Leela strictly better?
>
> The weaker Leela very occasionally does better than the strong version “by luck.” For example, there could be a move that looks good to the strong Leela but is, in fact, very bad for a subtle reason that even the strong model misses. If the reasons in favor of this move are themselves somewhat subtle, then the weak model might not even consider it, and just play a mediocre move, rather than the strong model’s bad move. But apart from such edge cases, the difference in playing strength is quite large, and we would be surprised if there are any classes of states where the weak model is systematically stronger.
>
> > how has the weaker Leela been trained? Is it plausible that, e.g. stronger MCTS during training of the weaker Leela means look-ahead is much less important, hence why it does not develop look-ahead (or does look-ahead only develop at sufficiently large and strong networks)?
>
> Both models were trained using supervised learning on MCTS rollouts (so the loss functions are the same as in MCTS training, but the data comes from an existing strong network, rather than the network under training). Given that MCTS trains the network to predict the results of tree rollouts, we don’t think strong MCTS would make look-ahead less important (this would only be true if the network was optimized end-to-end to maximize the playing strength of the overall MCTS process). We’d indeed guess that look-ahead becomes more prevalent for stronger networks, but of course, this would require future work to actually test.
>
> > The probe accuracy in Fig 8 is relatively high in early layers - how does this fit with the activation patching results that show the largest effect only in medium-to-late layers 6-11?
>
> Good question, we are not entirely sure what the answer is. One possibility is that there is a collection of different mechanisms involved in look-ahead, and not all of them activate in the same cases or are placed in the same layers. Probes might learn to exploit look-ahead mechanisms that are present in early layers, whereas the more targeted activation patching experiments only pick up on a subset of look-ahead mechanisms. L12H12 is already an example of a mechanism that is sometimes involved in look-ahead but is important less often than look-ahead mechanisms in general (such as the piece movement heads, contrast figs. 5 and 7). So it’s plausible that there are also some mechanisms that *only* probing picks up on.
>
> > As far as I am aware the precise details about the training process of this variant of Leela have not been published (in an academic venue). While this is beyond the scope of this paper, having these details (or as many as possible), e.g, in the appendix, would greatly help the scientific community to use this version of Leela as something that can be reliably reproduced.
>
> Indeed, most information about Leela is only available on the Leela Discord (which is public but naturally not designed to convey all that information compactly). We’d be happy to extend appendix A with some additional information (but a full description of Leela’s training details would fill its own paper).

---

> ### Comment · Reviewer_ScDd · 2024-08-12
> **Thank you for the detailed responses**
>
> Thank you for answering my clarifying questions and commenting on my (mostly optional) suggestions for improvement. I am positively surprised to see a full analysis of the value-network too. Overall I am happy with the authors' responses and additional results. I stand by my original score - I think this is a great interpretability case study with interesting results, that is ready for publication, well executed, and interesting to the NeurIPS audience.
>
> I have also read the other reviewers' comments and authors' responses and consider all raised issues sufficiently addressed (though I will happily take into account reviewers' future comments in case they disagree). abVe and p7Jw seem to be mainly concerned by the dataset filtering and only showing results on this filtered dataset - while this is understandable criticism at first, I think constructing such a dataset in a reasonable manner is a contribution in itself. The difficulty is that a large neural action/value predictor behaves quite differently from a search-based chess algorithm; in many board states it may be that the neural system does not use look-ahead-search but instead relies on memorization and exploitation of (statistically) similar patterns. It is thus crucial to first identify states where look-ahead-search may be needed. I am also not too worried about the use of the puzzle dataset - previous works (e.g. the 'Grandmaster-level chess without search' that abVe mentioned) have found strong correlation between puzzle performance and actual game-playing strength across a variety of models. The paper claims (and is very clear about this) that convincing evidence of look-ahead-search can be found *in these situations*. It does not claim that this is the main mechanism that the network uses at all times. Without proper filtering, the evidence for this mechanism might quickly "drown" within the (potentially large) number of situations where no look-ahead search is performed. While it would be interesting to know "how often" the network uses its learned search, this question is beyond the scope of the paper, whose goal is to clearly establish that the paper has learned to use look-ahead search at least sometimes (which I consider a highly non-trivial result).

---

> > ### Author Response · Authors · 2024-08-12
> >
> > Thank you for your positive feedback about our additional results and taking the time to write detailed and insightful responses to our rebuttal and the overall discussion on our work so far.

---

### Official Review · Reviewer_SpQ5 · 2024-07-17

**Soundness:** 4
**Presentation:** 4
**Contribution:** 3
**Rating:** 6
**Confidence:** 5

**Summary:**

This paper conducts a mechanistic interpretability analysis of Leela and shows evidence that it learns to look ahead in the network.

**Strengths:**

This paper provides a convincing answer to an important question: are networks like Leela's analogues of System 1-style pure intuition, or do they encode some amount of System 2-style calculation? The technique of corrupting the state and using activation patching is clever and effective. It's clear from the experiments that yes, there is some look-ahead being done. The authors are also careful to not overstate their claims: an actual broad search isn't necessarily being done, but at least some 1- or 2-step look-ahead is happening.

**Weaknesses:**

The paper is essentially an existence proof of lookahead, which is a useful contribution, but it would be nice to know something about the the prevalence of lookahead. The dataset is chosen to maximize the probability that lookahead is happening. How often does it happen?

I thought the claim that "look-ahead or other sophisticated algorithms should pick up on the importance of this difference, but shallow heuristics should mostly ignore it." required more justification. Some of the tactical patterns picked up by the method are so common that it's conceivable that a "shallow heuristic" is designed for them, rather than look-ahead.

The lack of results/explanations regarding why the squares of the 2nd move aren't important weakens the argument somewhat.

Why aren't there pawn or king heads, similar to the other piece types?

Nitpick: the motivation in the first paragraph about chess being different from other domains because in other domains "models can solve their tasks with a single simple algorithm" also applies to chess.

**Questions:**

How often does look-ahead occur?

What is the evidence that look-ahead is occurring as opposed to a shallow heuristic designed to pick up on common tactical patterns?

Why aren't the squares of the 2nd move similarly important?

Why aren't there pawn or king heads, similar to the other piece types?

**Limitations:**

Yes

---

> ### Author Rebuttal · Authors · 2024-08-06
>
> Thank you for your thoughtful feedback and questions! We answer them below and clarify potential misunderstandings.
>
> > the claim that "look-ahead or other sophisticated algorithms should pick up on the importance of this difference, but shallow heuristics should mostly ignore it." required more justification. Some of the tactical patterns picked up by the method are so common that it's conceivable that a "shallow heuristic" is designed for them, rather than look-ahead.
>
> We’d like to clarify that this claim is not a load-bearing assumption for our results. The sentence you quote only motivates why we use corrupted inputs that are similar to clean inputs (in the sense that a weak model gives the same output): if we used corrupted inputs that differ too much from the clean ones, then too many network activations would differ and drown out the evidence of look-ahead.
>
> If our argument for look-ahead was behavioral, i.e., the fact that Leela can solve these supposedly difficult puzzles, then it would indeed be an issue if some of them are also solvable by shallow heuristics. But our argument instead rests on observations about internal representations, and these observations suggest look-ahead irrespective of what exactly the corrupted inputs are. For example, if we could show that future move target squares are unusually important under random corruptions, this would be just as convincing in our mind.
>
> In summary, we agree that shallow heuristics could likely pick up on some of the tactics, but we don’t think this weakens our arguments.
>
> > What is the evidence that look-ahead is occurring as opposed to a shallow heuristic designed to pick up on common tactical patterns?
>
> We think all three lines of evidence we present favor look-ahead over shallow heuristics. For example, a shallow heuristic (which decides on the immediate next move based on matching to common patterns) would not need to explicitly represent which future moves will actually be played. But we find in section 3.3 that such representations of future moves exist, as a look-ahead hypothesis would predict. We similarly think that hypotheses that don’t involve look-ahead would not predict or explain any of our other results, but please let us know if you have specific concerns about this.
>
> > How often does look-ahead occur?
>
> Good question, but unfortunately difficult to answer precisely, for two reasons:
> - It depends on where we’d draw the boundary for “look-ahead” (for example, which effect size would be sufficient in our various experiments). While we think our results are strong enough to conclude that look-ahead is involved in at least some inputs (those with high effect sizes), it’s much less clear where exactly to set the threshold.
> - Our methods can only establish a lower bound on how often look-ahead occurs. For example, throughout the paper, we consider representations on future move target squares. In principle, additional look-ahead mechanisms that use entirely different pathways could exist, which we might not pick up on.
>
> With those caveats in mind: Our dataset is 2.5% of the initial Lichess dataset (see lines 83 and 91). On this dataset, we think look-ahead is involved in most states (but as discussed, it depends on where we draw the boundary). However, these 2.5% of states were, of course, not directly selected for high effect sizes; rather, we only used a behavior-based filtering procedure. So the true number of states where look-ahead is important is likely significantly larger than this among Lichess puzzles.
>
> > Why aren't the squares of the 2nd move similarly important?
>
> As discussed in line 163: “We are unsure why the squares of the 2nd move aren’t similarly important. This may simply be because the opponent’s move is typically “obvious” in our dataset or because suppressing the opponent’s best response doesn’t reduce the quality of the 1st move.”
>
> If the two hypotheses we give there are right, the apparent unimportance of the 2nd move would largely be an artifact of our experimental setup (studying tactics puzzles where the current player is winning). But we can’t rule that there is some interesting deeper reason that would require a more detailed understanding of Leela’s internal mechanisms.
>
> > Why aren't there pawn or king heads, similar to the other piece types?
>
> Great question! There are, in fact, pawn and king heads, they just involve a few complications, and so we decided to ignore them for simplicity. But we see that not mentioning them was a mistake, thank you for drawing attention to that! We’ll rectify this by adding a few comments to the paper:
> - A few king heads exist (based on attention patterns). But our dataset doesn’t contain many puzzles where the starting player moves their king on the first move, so we wouldn’t have much data for our piece movement head ablation. This is simply because king moves are much rarer in tactics puzzles.
> - Pawns move in different ways depending on context—they usually take one step forward, but they capture diagonally, and they can take two steps at once if they’re in their starting position. Based on attention patterns, it seems that there are distinct heads for these different types of movement, so we decided to ignore pawns to simplify the presentation in the paper. Again, we recognize this omission is also liable to lead to confusion, so we will bring this up in the paper.
>
> > Nitpick: the motivation in the first paragraph about chess being different from other domains because in other domains "models can solve their tasks with a single simple algorithm" also applies to chess.
>
> Thank you for pointing this out, our phrasing here is unfortunate and we’ll improve it. What we meant to gesture at is simply the difference in complexity between playing chess well and e.g., finding the path from the root of a tree to a specific leaf (which is the task studied in the most similar prior work).

---

> > ### Comment · Reviewer_SpQ5 · 2024-08-09
> >
> > Thanks for your detailed responses.

---

### Official Review · Reviewer_mWPe · 2024-07-17

**Soundness:** 3
**Presentation:** 4
**Contribution:** 4
**Rating:** 8
**Confidence:** 4

**Summary:**

This paper closely examines, using various forms of activation patching and probes, the inner workings of the state-of-the-art policy network of Leela Chess Zero (for the game of Chess). The authors find several different pieces of evidence that suggest it is likely that the network has learned to carry out some form of look-ahead search, at least in complex states that require this to arrive at the correct solution.

**Strengths:**

I found this to be a very interesting paper, and have little to remark on it. It's always good to see a paper that is actually trying to improve our understanding (and doing a good job at that), rather than just being "here is our new technique and it has bigger numbers". It is well written, using good examples and illustrations.

**Weaknesses:**

My only "important" criticism is that I object to the use of "existence proof" in both the abstract and the conclusion. I think this is too strong, and would suggest rephrasing to "evidence". I do find the evidence provided in the paper to be compelling, but still do not think it can be definitively described as proof rather than evidence.

---

Minor nitpicky comments:
- line 86 has a random period in the middle of a sentence (right before the footnote)
- I feel like it would be more natural for the paragraph of lines 222-224 to be moved a bit earlier. It's essentially describing a part of the experiment setup, but after some of the results have already been presented and discussed.
- Line 225: "are seem"

**Questions:**

1) Probably a difficult question, but curious to hear your thoughts if you have any: is it definitively possible to draw a hard line between "heuristics" and look-ahead? Especially depth-limited look-ahead (like what you seem to be finding evidence for in this paper)?

Here's what I'm thinking. If you only have a very basic heuristic (say, only material count), then you need at least 1 ply on search on top of that to find which moves let you capture something. But you could just add a smarter heuristic that counts the values of pieces your pieces can attack in one move, and then you no longer need one ply for this. With the most basic heuristic, you would need at least 2 plies of search to also consider the responses that the opponent can make. But, you could add a heuristic that counts which pieces defend their own friendly pieces, and that could partially account for what the second ply would typically be searching. Is there a limit to this? Could we conceivably create heuristics that detect the 3-ply situations explored in this paper, without search?

2) As a philosophical follow-up to the above: can we ever really distinguish between look-ahead search and heuristics, if the look-ahead search is depth-limited (I suppose any "learned look-ahead" in non-recurrent neural networks would always be depth-limited)?

**Limitations:**

There is a good discussion of limitations.

---

> ### Author Rebuttal · Authors · 2024-08-06
>
> Thank you for your review and insightful questions!
>
> We agree that “existence proof” was poor wording on our part and will rephrase that to “evidence” in both places, thank you for the suggestion. Thanks also for the minor comments, which we’ll incorporate.
>
> > is it definitively possible to draw a hard line between "heuristics" and look-ahead?
>
> This is a great question, and as you say, a full answer is probably a difficult philosophical problem. We give a few initial thoughts below.
>
> Drawing a very precise line around “heuristics” might not be possible (in the sense that the space between ad-hoc heuristics and general reasoning algorithms might be relatively continuous and it’s not exactly clear where the boundary is). Even so, we can probably say that certain algorithms do involve look-ahead and certain others don’t (while leaving the question open for some borderline cases).
>
> Implementing look-ahead by simply counting attacks/defenses in the current state, as you describe, is an interesting example of blurring this boundary. But capturing all relevant considerations this way likely only works well for 1- or perhaps 2-ply look-ahead. In the 3-ply case (which the paper focuses on), a piece might be moved twice (and already in the 2-ply case, heuristics would need to be very careful about counting defenders correctly to account for lines closing or opening up as pieces are moved).
>
> So while in some sense, there is no limit to what heuristics can implement—in the extreme case, a look-up table could implement arbitrary policies—we expect algorithms that explicitly use look-ahead to be far more efficient.
>
> > can we ever really distinguish between look-ahead search and heuristics, if the look-ahead search is depth-limited
>
> The thoughts outlined above suggest that a depth limit does not preclude a distinction between simple heuristics and look-ahead: while given enough capacity, both might be able to implement the same policies, they can differ in how much capacity that takes (and for sufficient depths, avoiding explicit look-ahead might be infeasible).
>
> Of course, our paper does not rest on these speculative conceptual claims; we aim to give evidence of explicit look-ahead rather than argue for its inevitability. (As one simple example, the method of counting attacks/defenses would not need to explicitly represent which future moves will actually be taken, so it would not predict our probing results.)

---

> > ### Comment · Reviewer_mWPe · 2024-08-09
> >
> > I acknowledge that I have read the rebuttal, and thank the authors for still taking the time to engage in an interesting discussion when there were also other reviewers' comments present for which it was probably more urgent to respond.
> >
> > I do not plan to further raise my score (which was already very high), but must say I am very puzzled by some of the perceived "weaknesses" described in some other reviews. I will argue against them in the Reviewer-AC discussion if necessary.

---

### Official Review · Reviewer_abVe · 2024-07-23

**Soundness:** 3
**Presentation:** 2
**Contribution:** 2
**Rating:** 4
**Confidence:** 4

**Summary:**

The paper try to discover and analyze the evidence of learned look-ahead in Chess playing network. They take experiments on filtered Chess puzzle dataset and the policy network of Leela chess policy. The claims are mainly analyzed from three perspectives, including Activation patching, attention head analysis and probing experiments.

**Strengths:**

1. The finding of the paper is interesting and inspiring. I like the idea of getting inside the network to understand how the policy network or transformer work to predict the optimal action.
2. The paper is easy to follow and well-written.
3. Experiments on different perspectives to help verify the conclusion.

**Weaknesses:**

1. The analysis seems to strongly related with the specific setting: Chess and transformer -- where each grid can be considered as a token. This can somehow limit the generality of the analysis method.

2. The main issue about the weakness comes from the experiments.

Since this is mainly a paper for discovering phenomenon and analysis, with no newly proposed algorithm or showing how this finding can help anything, the standard for NeurIPS paper requires it to have comprehensive experiments to validate the solidness of the finding. I do like this new finding, but the experiment is not enough to make it solid -- for instance, here are several experiments I think necessary for authors to present:

2.1. Present how this phenomenon happens without explicit designing test/evaluation dataset. The authors do a lot of work in section 2.2 on filtering the testing samples to make it more likely to have this phenomenon -- and they use this to claim that the policy network has this phenomenon. This is quite weird since you also somehow optimize the evaluation dataset to help you find the claimed phenomenon. If this is not a general finding or conclusion, it can make the argument pretty weak and may only work on some human created settings -- an extreme case is that I can manually design some test cases that can definitely present this ability but it lacks generality. I agree with the argument that not all boards present this phenomenon -- but at least the authors need to present more things, like in which condition this phenomenon is likely to happen and which will not, and how/why this can happen. Is there any key issue that prevent or support the emergence of this phenomenon? Also the authors filter out all puzzles that Leela Chess may fail. This is really weird and can largely weaken the conclusion since if you have already tested it to solve the puzzle, it is quite normal that the representation of policy network can have something closely related with future optimal action (in the activation patching experiments or probing one).

2.2 Is this conclusion generalizable on other transformer-based Chess policy? For example in "Grandmaster-Level Chess Without Search
" they also have a policy trained by supervised learning with no RL or MCTS like the leela chess does. Will this setting affect the emergence of this behavior?

2.3 Does only the policy network have this phenomenon or the same thing can happen at critic network? For example you can test it on leela-chess/stockfish evaluation function.

There are also a lot of ablation studies on modules or hyperparameters to make the conclusion solid. So again, solid and comprehensive experiments and ablation studies are necessary for such analysis/discovering paper, while this paper doesn't offer enough evidence.

**Questions:**

1. There are several papers about transformer/Chess/AlphaZero that are also related work to this paper, including:
(1) Zahavy, Tom, et al. "Diversifying ai: Towards creative chess with alphazero." arXiv preprint arXiv:2308.09175 (2023).
(2) Feng, Xidong, et al. "Chessgpt: Bridging policy learning and language modeling." Advances in Neural Information Processing Systems 36 (2024).
(3) Noever, David, Matt Ciolino, and Josh Kalin. "The chess transformer: Mastering play using generative language models." arXiv preprint arXiv:2008.04057 (2020).
(4) Stöckl, Andreas. "Watching a language model learning chess." Proceedings of the International Conference on Recent Advances in Natural Language Processing (RANLP 2021). 2021.

2. Can the author explain more on the activation patching. Is it something like: you replace one neuron in one layer from the neuron of the same place in the forward process of corrupted board? Then I am wondering how you choose this corrupted board. Also in figure 3 why 1st/3rd but no 2nd action?

3. Also I 'd suggest authors to include more illustrative examples/figures in the main paper or appendix to help readers (and those not familiar with chess) to understand the phenomenon.

**Limitations:**

See weaknesses.

---

> ### Author Rebuttal · Authors · 2024-08-06
>
> Thank you for your thorough review and suggestions!
>
> **Policy vs value network (2.3):** Great suggestion, thank you! We have now run all our experiments on Leela’s value head and will include these in the paper. We used the logs odds of the win probability as the effect size for patching experiments. (Probing experiments operate only on the shared network body so don’t differ between heads.) All our findings for the policy head still hold for the value head, see the PDF attached to our general response. The only new finding is that L14H3 seems very important to the value head (Fig. 2 in the attached PDF), but L12H12 is still just as important as it was for the policy head (log odds reduction of 0.49 when we ablate it).
>
> **On dataset filtering (2.1):** We appreciate that filtering our dataset can seem strange. However, we think this filtering process is essential to test our hypothesis and doesn’t negatively affect the validity of our results. We clear up potential misunderstandings below.
>
> > This is quite weird since you also somehow optimize the evaluation dataset to help you find the claimed phenomenon.
>
> Investigating our claimed phenomenon fundamentally requires creating a suitable evaluation dataset *somehow*. As discussed (e.g. line 80), our claim is only that Leela often uses look-ahead *in certain types of states*, namely tactically complex ones. Such hypotheses specific to a certain input distribution are omnipresent in mechanistic interpretability (e.g. [1, 2, 3, 4]). To test this hypothesis, we naturally evaluate it on those states for which we claim Leela uses look-ahead. Our filtering process formalizes the vague notion of “tactically complex states.”
>
> > may only work on some human created settings -- an extreme case is that I can manually design some test cases that can definitely present this ability but it lacks generality
>
> Indeed, manually creating inputs would be very suspect, so we want to emphasize that we start with an existing dataset of inputs and then apply a simple automated filtering step. Importantly, we also do not use internal representations in any way during filtering.
>
> > Also the authors filter out all puzzles that Leela Chess may fail. This is really weird and can largely weaken the conclusion
>
> We focus on puzzles that Leela solves simply so we can apply our interpretability methods. All our methods check whether Leela represents a specific future line. In correctly solved puzzles, we can look for representations of the *correct* continuation. In puzzles that Leela fails to solve, this wouldn’t tell us as much: Leela may be representing an incorrect continuation (and hence fail the puzzle), but there are many incorrect lines (vs only one correct one). So we can’t test for look-ahead like we can for correctly solved puzzles. We briefly explain this in line 89 but realize that this deserves more space, so we’ll incorporate this explanation in the updated paper.
>
> As to whether this weakens the conclusion: we want to explain why Leela often solves puzzles correctly and don’t claim to explain why it sometimes fails. We’ve tried to be transparent about this (e.g. line 11 in the abstract) but will do another editing pass to make this clear.
>
> > if you have already tested it to solve the puzzle, it is quite normal that the representation of policy network can have something closely related with future optimal action
>
> Crucially, we don’t think that Leela solving a puzzle necessarily implies the claims we make about internally represented look-ahead. Drawing such mechanistic conclusions from behavioral evidence is dubious, which is why interpretability is needed in the first place. We agree that the model’s ability to solve difficult puzzles might *lead us to expect* that its representations would have something to do with future actions. Our paper's contribution lies in actually testing a more precise version of this hypothesis.
>
> **Testing other transformer-based chess-playing networks (2.2):** This would be an excellent future direction! Unfortunately, the model weights you mention were released only a bit over a month before the NeurIPS deadline. Transferring our experiments to new models would be a non-trivial effort, e.g., since we need to add instrumentation for our patching experiments.
>
> Regarding RL vs. supervised learning, note that the version of Leela we use was, in fact, trained using supervised learning (see line 422 in the appendix for details). The difference is just that it uses data from rollouts of an MCTS-trained Leela, rather than Stockfish evaluations.
>
> **Answers to questions:**
> 1. Thank you for the references, we will cite these as additional examples of transformer-based chess models.
> 2. That’s right, except we patch entire activation vectors on a square, rather than individual neurons. Our procedure for finding corrupted inputs is described in line 120 and appendix D. In brief, we randomly generate corruptions and then pick one that doesn’t change a weak model’s output, but does change Leela’s output a lot. In Fig. 3, we omit the 2nd move target since it is very often the same as the 1st move target (see line 93). Figure 9a demonstrates that the effect is mainly about the 1st and 3rd rather than 2nd move, which is why show the 1st instead of 2nd move.
> 3. Are there specific aspects that you think are difficult to follow and would benefit from additional figures? That would help us a lot!
>
> [1] Wang et al, 2022. Interpretability in the Wild: a Circuit for Indirect Object Identification in GPT-2 small
>
> [2] Hanna et al, 2023. How does GPT-2 compute greater-than?: Interpreting mathematical abilities in a pre-trained language model
>
> [3] Lieberum et al, 2023. Does Circuit Analysis Interpretability Scale? Evidence from Multiple Choice Capabilities in Chinchilla
>
> [4] Nanda et al, 2023. Fact Finding: Attempting to Reverse-Engineer Factual Recall on the Neuron Level

---

> > ### Comment · Reviewer_abVe · 2024-08-13
> >
> > Thank you for your rebuttal! I am satisfied with additional explanations and results.
> >
> > But I am still concerned with the dataset filtering -- I appreciate the authors' transparency about this. But if this is a conclusion that only applies for 20k puzzle boards this can largely weaken my surpriseness about the conclusion, especially that the title seems to ignore this limited scope.
> >
> > I will keep my score for now -- but maybe will increase it after my discussion with other reviewers and ACs. Thanks for your engagement again.

---

> > > ### Author Response · Authors · 2024-08-13
> > >
> > > Thank you for your response! We appreciate the time to read through our rebuttal.
> > >
> > > > if this is a conclusion that only applies for 20k puzzle boards
> > >
> > > We wanted to mention that the lichess puzzles come from real games played on the lichess website. These games were analyzed via Stockfish at 40 meganodes (which "took more than 50 years of CPU time" by the lichess team), and we further filtered these down using a simple filtering method (discarding board positions that a smaller model can solve). Therefore, our dataset is a subset of board positions from real games filtered for our description of hard positions that might require look-ahead to solve.
> > >
> > > Thank you for engaging with our discussions again, we really appreciate your responses!

---

### Author Rebuttal · Authors · 2024-08-06

Thank you for your detailed reviews and suggestions! We’re encouraged that you found the results *“interesting and inspiring”* (abVe), thought the paper gave a *“convincing answer to an important question”* (SpQ5), and said it has *“great experimental work, with convincing evidence, important controls, and multiple investigations aimed at the same question”* (ScDd).

**Value head experiments:** In response to feedback from reviewers abVe and ScDd, we’ve added versions of all our experiments that use Leela’s value head instead of its policy head to measure the effect of interventions. The results are similar to the policy head results in the paper (likely because most of the network is a shared body). We’ve attached the new results as a PDF, the interpretation of all figures is exactly the same as for the corresponding figures in the paper. The only difference is that we use the log odds of the win probability rather than the log odds assigned to the best move. Note that the probing results (section 3.3) are not specific to value or policy head, since they operate entirely on the shared body.

We’re also making a few minor changes for clarity based on feedback from reviewers. We mention these in response to individual reviews, along with answering questions and clarifying a few misconceptions.

---

### Decision · Program_Chairs · 2024-09-25

**Decision:**

Accept (poster)

**Comment:**

The paper investigates the presence of look-ahead the neural network of Leela. It is fairly difficult to provide reliable evidence either way, but the authors do offer a sensible approach. There is a disagreement amongst the reviewers about how valid the set-up is, and what would be the future impact, but this is not surprising given the unorthodox problem. Considering how much engagement the paper generated from the reviewers, and the strong positive reaction from a subset of them, I feel that the paper could be of interest for the audience of the conference. I would encourage the authors to take into account the comments of the reviewers, especially in clarifying some of the details and claims.